# A dataset for lake level changes in the Tibetan Plateau from 2002 to 2021 using multi-altimeter data

Jiaming Chen[1,3], Jingjuan Liao[1, 2*,] Yanhan Lou[1,4], Shanmu Ma [1,4], Guozhuang Shen[1,2], Lianchong Zhang[1]

[1]Key Laboratory of Digital Earth Science, Aerospace Information Research Institute, Chinese Academy of Sciences, Beijing, 100094, China.

[2] International Research Center of Big Data for Sustainable Development Goals, Beijing 100094, China.

[3] Institute of Geodesy and Geoinformation, University of Bonn, Bonn, Germany.

[4] University of Chinese Academy of Sciences, Beijing 100049, China.

*Correspondence*: Jingjuan Liao (liaojj@aircas.ac.cn )

**Abstract.**

The Tibet Plateau (TP), known as the Roof of the World and the Water Tower of Asia, has the largest number of lakes in the world, and because of its high altitude and near absence of disturbances by human activity, the plateau has long been an important site for studying global climate change. Hydrological stations cannot be readily set up in this region, and *in situ* gauge data are not always publicly accessible. Satellite radar altimetry has become a very important alternative to *in situ* observations as a source of data. Estimation of the water levels of lakes via radar altimetry is often limited by temporal and spatial coverage, and, therefore, multi-altimeter data are often used to monitor lake levels. Restricted by the accuracy of waveform processing and the interval period between different altimetry missions, the accuracy and the sampling frequency of the water level series are typically low. By processing and merging data from eight different altimetry missions (Envisat, ICESat-1, CryoSat-2, Jason-1, Jason-2, Jason-3, SARAL, and Sentinel-3A), the developed datasets provided the water level changes for 361 lakes (larger than 10 km$^2$) in the TP from 2002 to 2021 (194 lakes for the time series from 2002 to 2021 and 167 lakes for the time series from 2010 to 2021). The period for the lake level change series, which affords high accuracy, can be much longer for many lake systems. The present datasets and associated approaches are valuable for calculating the changes in lake storage, trend analyses of the lake levels, short-term monitoring of the overflow of lakes, flooding disasters on the plateau, and the relationships between changes in the lake ecosystems and changes in the water resources.

## 1 Introduction

As primary water reservoirs, lakes not only play an important role in the supply and adjustment of surface water but also reflect the impact of climate change and human activities on regional and global environmental change (Adrian et al., 2009; Schindler, 2009; Song et al., 2015). The water level of lakes is a key indicator for regional climate change and human disturbance. Generally, it is assumed that the changes in lake bottoms are very slight over decades, so understanding the changes in lake levels can help to evaluate the impact of climate change and human activities on regional water resources.

Observation by use of a water gauge is the traditional method to measure the changes in water levels in lakes; *in situ* gauge measurement of lakes can afford high precision but such equipment is expensive to maintain and challenging to operate in remote areas. Furthermore, the total number of monitoring stations has decreased in recent years (Frappart et al., 2006; Kleinherenbrink et al., 2014), and lake level data in many countries and regions are not freely available to the public. Alternatively, satellite altimetry technology is an effective tool that can be used to measure the dynamics of the surface elevation of the Earth and has been successful in measuring lake levels. The Tibetan Plateau (TP), known as the Roof of the World and the Water Tower of Asia, has numerous and some of the largest natural lakes in the world, and because of its high altitude and the near absence of human disturbances, the plateau is an important location for studying global change. Changes in the water level in lakes are one of the important indicators for the water balance of the TP and these are directly affected by

temperature, precipitation, evaporation, glaciers, perennial snow cover, and permafrost (Zhang et al., 2012; 2013a; 2013b). The TP is the source of many major rivers, and more than 1.4 billion people depend on water resources from the plateau (Pritchard, 2017). However, due to the vastness and remoteness, it is a challenge to set up *in situ* monitoring stations. There are only a few lakes (such as Qinghai Lake, Namtso, and Yamdrok Yumtso) with *in situ* gauge stations for lake level measurements (Zhang, 2018). Most lakes in the TP lack such a measurement capability making it difficult to understand the long-term spatial and temporal characteristics regarding the evolution and dynamics of the water levels of the lakes.

Satellite altimetry has become the most important means to measure lake levels and their changes in the plateau. Numerous studies have focused on the use of satellite altimeters for measuring changes in lake levels in the TP. For example, Gao et al. (2013) employed multi-altimeter data from Envisat, CryoSat-2, Jason-1, and Jason-2 to examine water level changes at 51 lakes between 2002 and 2012 in the OTP. Zhang et al. (2011) used Ice, Cloud, and the land Elevation Satellite (ICESat) data to determine changes in lake levels in Tibet from 2003 to 2009. Hwang et al.(2016) obtained two decades of lake level measurements at 23 lakes in the TP from the T/P-family altimeters. Song et al. (2015) combined ICESat-1 and Cryosat-2 altimetry data to access the water level dynamics of Tibetan lakes from 2003 to 2014. Kleinherenbrink et al. (2015) and Jiang et al. (2017) used the CryoSat-2 data to measure changes in the water levels at 125 lakes and 70 lakes in the TP, respectively. Hwang et al. (2019) constructed a lake level time series for 61 lakes on the Tibetan Plateau between 2003 and 2016 and discussed the trends of the time series. Li et al. (2019) constructed high-temporal-resolution water level datasets for 52 large lakes on the Tibetan Plateau. These studies in the TP reveal that estimation of the lake levels with a given radar altimeter is often limited by temporal and spatial coverage, and, therefore, multiple altimeters are needed to obtain multiple decades of changes in the water levels of lakes. Although some websites also provide open access lake level data in the TP, the number of lakes is limited, e.g., Hydroweb has only 36 lakes and DAHITI has only 62 lakes in the TP (Cretaux et al. 2011; Schwatke et al. 2015). However, due to the large size of the radar altimeter footprint and contaminations from the steep lakeshore or surrounding land, the observations of lake levels via satellites are noisy, and it is difficult to obtain the distance from the altimeter to the nadir points. Therefore, waveform retracking processing may be used to remove the contamination by land signals when lake levels are retrieved from multi-altimeter data. In this study, by combining eight sets of altimeter data from Envisat, ICESat-1, CryoSat-2, Jason-1, Jason-2, Jason-3, SARAL, and Sentinel-3A, the trends of the changes in the water levels for 361 lakes ($>10$ km$^2$) in the TP during 2002-2021 were estimated using retracking and outlier detection algorithms. The primary objective of this study was to determine the changes in the water levels of 361 lakes in the TP from multi-altimeters and evaluate the accuracy of the time series and the performance of the multi-altimeter data with respect to monitoring the long-term variations in the water levels of the lakes. Readers can access the dataset described in this paper at https://doi.org/10.1594/PANGAEA.939427 (Chen et al., 2021), and comparison of our study with related previous studies is shown in Table 1

Table 1 Comparison of this study with previous studies

| Reference | No. of Lakes | Period | Data Source | Dataset Public or not |
|---|---|---|---|---|
| Jiang et al. (2017) | 70 | 2003-2015 | IceSat-1, Cryosat-2 | N |
| Zhang et al. (2017) | 68 | 1989-2015 | IceSat-1, Landsat | N |
| Li et al. (2017) | 167 | 2002-2012 | IceSat-1, Envisat | N |
| Hwang et al. (2019) | 59 | 2003-2016 | Jason-2/3, SARAL, IceSat-1, Cryosat-2 | N |
| Li et al. (2019) | 52 | 2000-2017 | Jason-1/2/3, Envisat, Cryosat-2, IceSat-1 | Y |
| Zhang et al. (2019) | 62 | 2003-2018 | IceSat-1/2 | Y |
| Hydroweb | 36 | 1993-2022 | ERS-2, Envisat, T/P, IceSat-1, SARAL, Jason-1/2/3, Cryosat-2, Sentinel-3A | Y |

| DAHITI | 62 | 2003-2022 | ERS-2, Envisat, SARAL, Sentinel-3A, Cryosat-2, IceSat-1, Jason-2/3, | Y |
| This Study | 361 | 2002-2021 | Envisat, SARAL, IceSat-1, Cryosat-2, Jason-1/2/3, Sentinel-3A | Y |

## 2 Study area and data

### 2.1 Study area

The TP is in the southwest of China and covers about 27% of the total area of China (Zhang et al., 2002), and its location and details are shown in Fig.1. There are more than 1000 lakes of >1 km$^2$ (Wan et al., 2016) in the TP, most of which belong to inland drainage systems. Based on coverage by altimeter data, 361 lakes of >10 km$^2$ in the TP were selected as the objects of study. Among these lakes, there were 13 lakes of > 500 km$^2$, 79 lakes of 100-500 km$^2$, 69 lakes of 50-100 km$^2$, and 200 lakes of 10-50 km$^2$. Most of these lakes are inland lakes with surface runoff, precipitation, snow and ice melting, springs, and

underground runoff as their main sources of water recharge. Due to minimal impact by human activity, changes in the water levels in the lakes in the region are driven mainly by natural factors such as precipitation and temperature, which are important indicators of changes in the regional climate and the ecological environment.

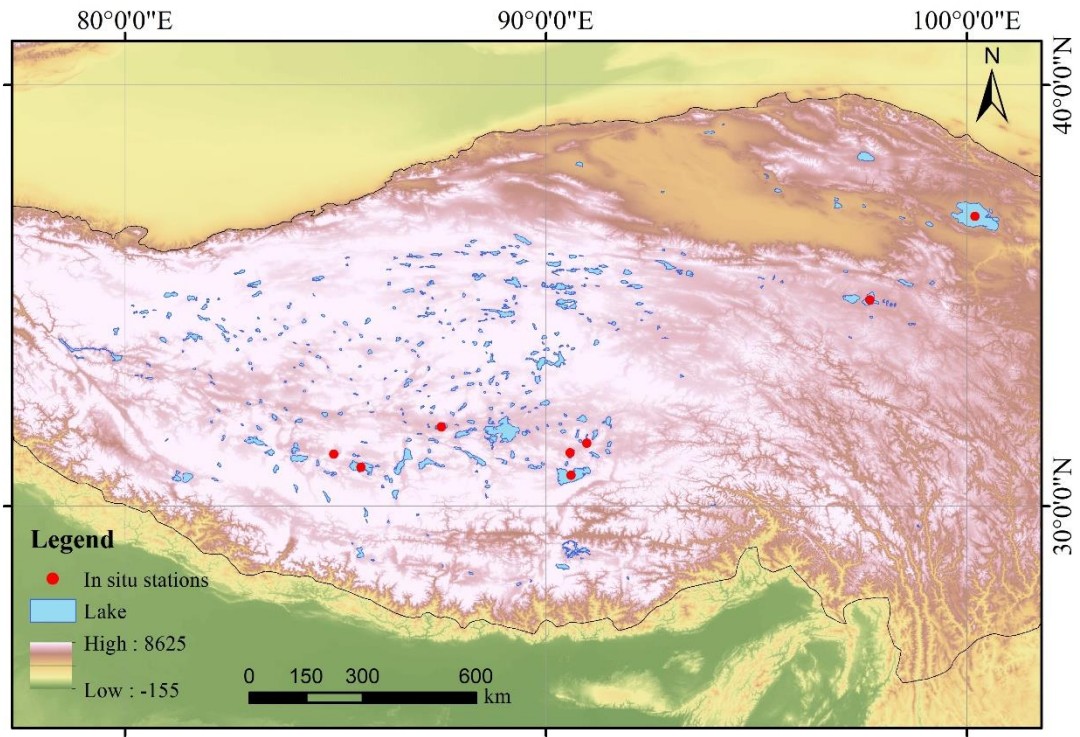

Fig 1. Location and distribution of lakes in the TP (The DEM of the base map is from the Global Multi-resolution Terrain

Elevation Data 2010(GMTED2010)(GMTED: https://topotools.cr.usgs.gov/gtmed_viewer/)

### 2.2 Data

#### 2.2.1 Multi-altimeter data

Eight sets of altimeter data from Envisat, ICESat-1, CryoSat-2, Jason-1, Jason-2, Jason-3, SARAL, and Sentinel-3A were used to extract the water levels of the lakes in the TP to obtain the lake level time series with high-space coverage. The details of

the multi-altimeter data are given in Table 2. Envisat, CryoSat-2, and Sentinel-3A data provided by the European Space

Agency (ESA) were available for 121, 352, and 106 lakes, respectively. Jason-1, Jason-2, and Jason-3 data provided by the Centre National d'Etudes Spatiales (CNES) and the National Aeronautics and Space Administration (NASA) were available for 48, 71 and 28 lakes, respectively, due to the relatively sparse ground tracks. Note that Jason-1/2 experience interlaced orbit (Jason-2 from Oct. 2016 to June 2017, Jason-1 after February 2009) which increasing the spatial coverage of Jason-1/2.

ICESat-1 data provided by NASA were available for 124 lakes. ICESat-1 is different from above radar altimeter, and its technique afforded high spatial resolution and smaller footprint. SARAL is a joint mission of the Indian Space Research Organization (ISRO) and CNES and is a continuation of the Envisat mission. SARAL data were available for 135 lakes in the TP.

Table 2 Details of the multi-altimeter data used in this study

| Mission | Sensor | Duration | No. of lakes | Repeat period(days) | Diameter of footprint (km) |
|---|---|---|---|---|---|
| Envisat | RA-2 | 2002.05-2012.04 | 121 | 35 | 1.7 |
| ICESat-1 | GLAS | 2003.02-2009.10 | 124 | 91 | 0.07 |
| CryoSat-2 | SIRAL | 2010.07-2021.07 | 352 | 369 (30d sub-cycle) | 1.6 (across), 0.3 (along) |
| Jason-1 | Poseidon-2 | 2002.01-2012.03 | 48 | 9.92 | 2.2 |
| Jason-2 | Poseidon-3 | 2009.12-2017.05 | 71 | 9.92 | 2.2 |
| Jason-3 | Poseidon-3B | 2016.02-2020.12 | 28 | 9.92 | 2.2 |
| SARAL | Altika | 2013.03-2016.05 | 135 | 35 | 4 |
| Sentinel-3A | SRAL | 2016.03-2019.09 | 106 | 27 | 2 (across), 0.25 (along) |

In addition, a dataset on the shapes of the lakes generated by Wan et al. (2016) was selected to determine whether the altimeter data encompassed the lakes, and a buffer of 1 km around the shape of the lake was generated to determine the change in the boundary of the lakes during the past 20 years.

**2.2.2 *In situ* data**

*In situ* data on eight lakes were used to validate reliable information on the lake level time series from the multi-altimeter data. Table 3 lists details of the *in situ* data on the eight lakes. The *in situ* data for Qinghai Lake and Ngoring Lake were from the Hydrology and Water Resources Survey Bureau in Qinghai Province and from the Yellow River Commission of the Ministry of Water Resources, respectively, and the *in situ* data on Bamco, Dagzeco, Dawaco, Namco, Pungco and Zhari Namco were

from the Institute of Tibetan Plateau Research, Chinese Academy of Sciences (Lei, 2018; Wang, 2018).

Table3 Details of the *in situ* data for eight lakes as used for validation

| Lake name | Date | Coordinates (°) | Reference | Mode[3] |
|---|---|---|---|---|
| Qinghai Lake | 2010.05-2019.09 | 100.20, 36.89[1] | 1985[2] | Absolution |
| Ngoring Lake | 2010.01-2015.12 | 97.70, 34.90 | 1985 | Absolution |
| Bamco | 2013.06-2017.10 | 90.58, 31.27 | Customize | Relative |
| Dagzeco | 2013.06-2016.10 | 87.52, 31.89 | Customize | Relative |
| Dawaco | 2013.06-2016.10 | 84.96, 31.24 | Customize | Relative |
| Namco | 2007.04-2016.12 | 90.60, 30.74 | Customize | Relative |
| Pungco | 2014.05-2017.10 | 90.97, 31.50 | Customize | Relative |
| Zhari Namco | 2012.12-2017.10 | 85.61, 30.93 | Customize | Relative |

[1] the first figure is longitude, the second figure is latitude

[2] 1985 indicates the 1985 national height datum of China

[3] Absolute mode is the elevation relative to the geoid, and relative mode is the elevation relative to the average value (set to 0) of the *in situ* data

## 3 Methods

### 3.1 Extraction of lake water levels

With respect to the extraction of the water level data from the satellite altimetry, there is uncertainty as to whether there is a
valid footprint falling on the lakes; this problem can be addressed by comparing the geographic coordinates of the footprints with the shape of the dynamic dataset for the lake. However, it would take considerable time to extract the dynamic shape file. A static shape dataset for the Tibetan Plateau was used in this study (Wan et al., 2016); we also generated a 1 km buffer for the shape to solve the situation regarding the changes in the boundary of lakes during the past 20 years. After picking out the available footprints, the height of the lake surface height can be calculated based on using Eq. (1) for each footprint:

$$H = Alt - (R_{range} + \Delta R_{dry} + \Delta R_{wet} + \Delta R_{iono} + \Delta R_{tide} + \Delta R_{correction}) - N_{geoid} \qquad (1)$$

where $Alt$ is the satellite altitude, $R_{range}$ is the distance between the altimeter and the lake surface, $\Delta R_{dry}$ is the dry troposphere, $\Delta R_{wet}$ is the wet troposphere, $\Delta R_{iono}$ is the ionospheric correction, $\Delta R_{tide}$ includes the solid earth tide, the pole tide, and the ocean tide corrections, $N_{geoid}$ is the geoid height with respect to the ellipsoid, for which the 2008 Earth Gravitational Model (EGM2008) was used in this study (Pavlis et al., 2012), and $\Delta R_{correction}$ stands for the retracking
correction $\Delta R_{retrack}$ for radar altimetry and the saturation correction $\Delta R_{saturation}$ for the laser altimetry. With the exception for $\Delta R_{retrack}$, all the corrections above are included in the altimetry data product.

### 3.1.1 Waveform retracking

The accurate measurement of the distance from the altimeter to the nadir points in inland water bodies poses a significant challenge due to the potential interference or submergence of waveforms by signals from adjacent land areas. Consequently,
the implementation of retracking correction is of great importance in mitigating the influence of land signals when utilizing radar altimetry data for inland water body studies (Martin et al., 1983; Lee et al., 2008). In this study, the automatic multiscale-based peak detection retracker (AMPDR) (Chen et al., 2021), which is suitable for Jason-2/3, Sentinel-3A/B and Cryosat-2, and can get good results (Chen and Liao, 2020). However, sometimes there were biases for the retracking correction caused by the hooking effect or the scatter signal of the off-nadir point for Jason-1, Envisat, and SARAL. Therefore, some
modifications for AMPDR were adopted for Jason-1, Envisat, and SARAL data in this study.

To ensure that the different typologies of multi-waveforms can be dealt with, we implemented a two-step process for the modified AMPDR here. The steps of the modified retracker are illustrated in Fig. 2. The optimal retracked range was determined using several criteria:

(1)    The optimal retracked levels should be within the range $H_{DEM} \pm 20$ m.
(2)    The DistanceThresh in AMPDR produced the smallest difference for the median of the water levels derived from the neighboring cycles if time has continuity.

(3)    The standard deviation of the water level of the current cycle was decreased if the data were not continuous; that is to say, the difference between the neighboring cycle and the current cycle was more than ten days or several months.

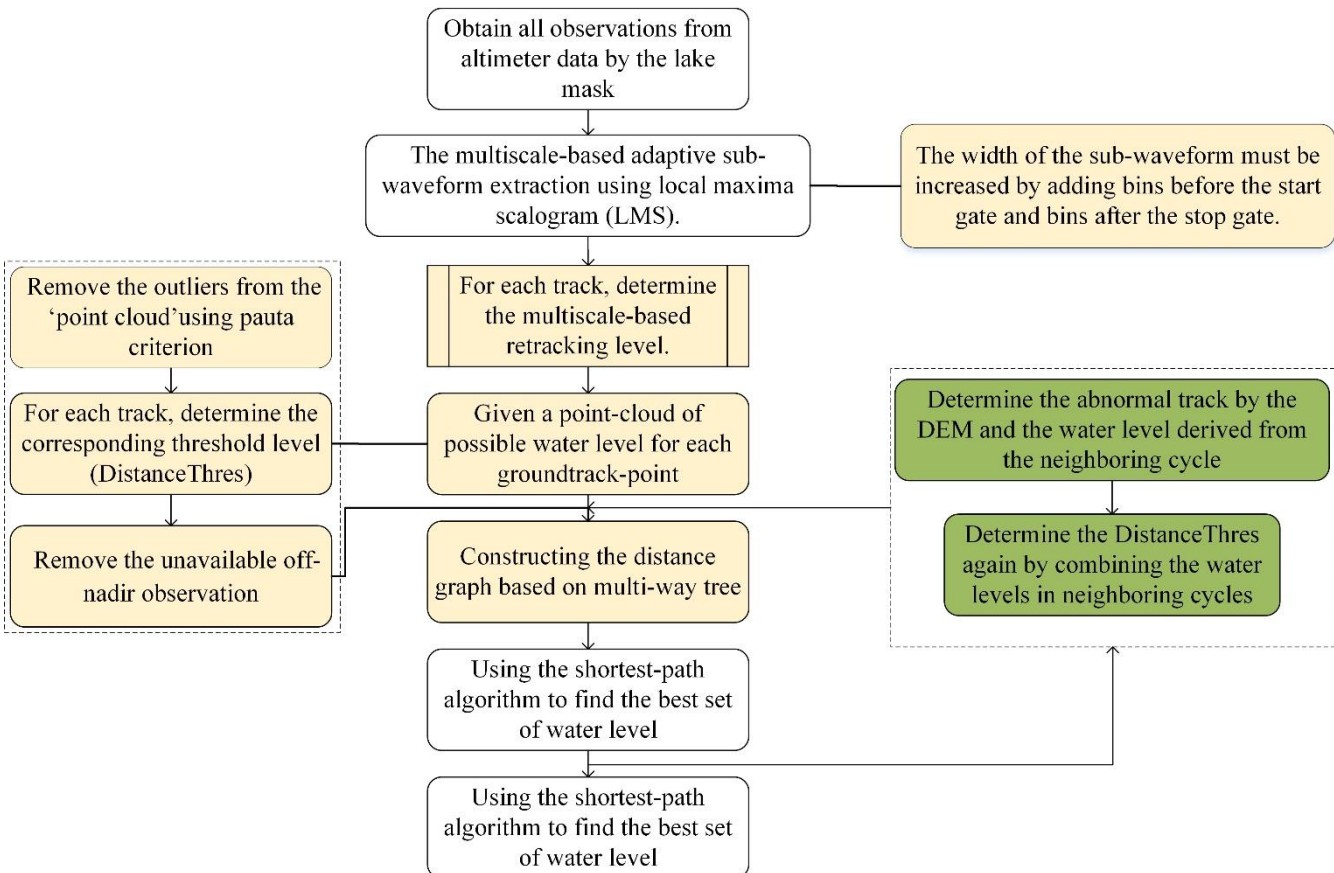

Fig. 2 Flowchart outlining the waveform retracking process. Steps with a yellow background are the preparation steps for using the shortest path algorithm. Steps with a green background are the retracking for the abnormal track by the selected DEM.

In the first run, the normal operation of the AMPDR was considered, and the lake level time series was calculated. Details regarding the definition and implementation of AMPDR are available elsewhere (Chen et al., 2021). Next, a second run of the retracking for the abnormal track, which was selected by the Digital Elevation Model (DEM) and the water level derived from the neighboring cycle, was implemented. However, this time the DistanceThresh in AMPDR was constructed by one of three minimum second-order difference quotients of the cumulative distribution function (CDF) of the rounded water levels. In this

way, it was ensured that the DistanceThresh was approached by the median of the water levels in neighboring cycles. Additionally, the retracking point from the ICE-1 algorithm was added to the construction of the "point cloud" and the CDF given that the multiscale-based adaptive threshold retracking would fail in some situations. An example of the operation of the modified two-step retracker is shown in Fig. 3.

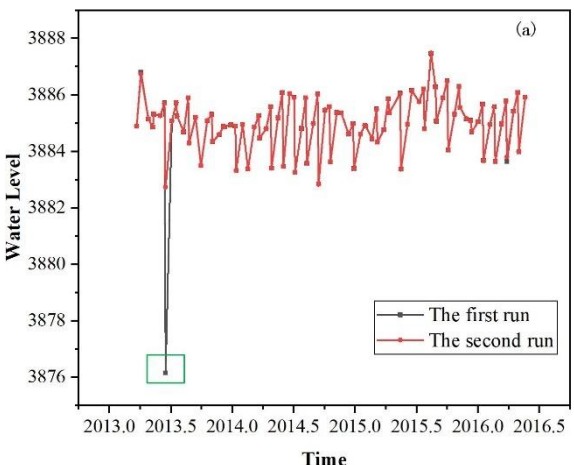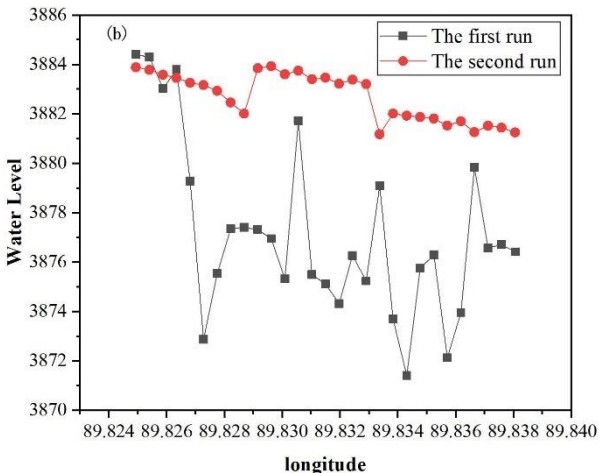

Fig. 3 An example of the operation of the modified two-step retracker. (a) shows the two water level time series for processing by the two-step retracker. (b) shows the along-track water level in the green rectangle from (a) when processing by the two-step retracker.

### 3.1.2 Removal of noise footprints

Due to the use of a 1 km buffer to pick out the shape of the available footprints, there would be many noise footprints caused by the reflected signals of the terrain or by the scatter signals of the off-nadir points. The noise footprints should be removed before constructing the lake level time series. Waveform classification is an effective method for identifying the noise footprints. Studies have proposed the use of various waveform classification methods and good recognition results have been achieved (Göttl et al., 2016; Lee et al., 2016; Marshall and Deng, 2016; Shen et al., 2017).

Different from the previous study whereby the waveforms are divided into multiple classes, this study only needs to divide the waveforms into noise and non-noise waveforms using a random forest (RF) classifier. The RF classifier was set up using a training set of approximately 300 waveforms over inland lakes for each altimetry, Additionally, the following features of the waveforms were selected: the pulse peakiness (Strawbridge and Laxon, 1994), the mean value of the waveform, the skewness of the waveform, the kurtosis of the waveform, the amplitude of the waveform, the width of the waveform determined by the Offset Center of Gravity (OCOG) retracker (Bamber 1994), the bin position corresponding to the center of gravity determined by the OCOG retracker, and the peakiness of the left and right pulse (Ricker et al., 2014). After discarding these noise footprints, the tracks with fewer than five observations were excluded from this study.

### 3.1.3 Construction of time series

Despite removing the noise footprints using waveform classification, the dataset also has outliers in the lake level time series for each cycle of a certain altimeter. Therefore, any point level in each cycle yielding a difference larger than three times the standard deviation ($3\sigma$ rule) was removed. Then, the lake level time series was estimated using the R package tsHydro (https://github.com/cavios/tshydro). The core of tsHydro is a state-space model consisting of a process model and an observation model, providing a robust time series for altimeter observations.

$$H_i^{true} = H_{i-1}^{true} + \sqrt{t_i - t_{i-1}}\,\sigma_{RW}z_i, \qquad z_i \sim N(0,1) \tag{2}$$

$$H_{ij}^{obs} = H_i^{true} + \sigma_{obs}\varepsilon_{ij} \tag{3}$$

The process model is used to describe the relationship between the true water level $H^{(true)}$, and the observation model is described by the observed water level $H^{(obs)}$, with an error term $\varepsilon_{ij}$, being used to describe the relationship between $H^{(obs)}$ and $H^{(true)}$. The scaling parameter $\sigma_{RW}$ is defined as the standard deviation of the random walk in a time step. The model is

described in detail by Nielsen et al. (2015). According to the Laplace estimation, the mean value of the range was selected to represent the water level of the lake for each cycle. Meanwhile, the standard deviation of each cycle was reserved to evaluate the uncertainty of the time series.

## 3.2 Fusion of multi-altimeter time series

It is not uncommon that the geoid between different altimeters should be different. Before merging the lake level from different altimeters, the geoid should be unified as WGS84/EGM 2008. The reference system of Jason-1/2/3 is the Topex/Poseidon (T/P) ellipsoid system instead of the WGS84 system, thus it was necessary to perform an ellipsoid system transformation from T/P to WGS84 by subtracting 0.71 m from the vertical height (Bhang et al., 2007).

Due to the variations in orbits and the disparities between instruments, systematic biases existed among the lake level time series extracted from the multi-altimetry, although they were corrected to the same reference system. In most studies (Li et al., 2019; Gao et al., 2013; Huang et al., 2016), the altimeters with the longest overlap period would be merged for the first time, but there may be some special situations whereby for some lakes the lake level time series for each altimeter cannot be merged. In this study, the dynamic reference time series was used to merge the lake-level time series. We first merged the two products with the longest period for the time series and chose the altimeter-derived water level with the longer time series as the baseline. Then systematic biases between another altimeter and the baseline will be removed by subtracting the mean discrepancy during the overlap period compared with the reference series (Lee et at., 2011; Kropáček et al., 2012) according to Eq. (4). Then, the same process was applied to the remaining products and the merged products connecting the three altimeters. The result for the merged altimetry data when all sensors are available is shown in Fig. 4a and 4b.

$$Series2_{cor}(t_i) = Series2_{ini}(t_i) + \left(\overline{Series1_{ref}} - \overline{Series2_{ini}}\right) \tag{4}$$

where $Series2_{ini}(t_i)$ is the uncorrected lake level at time $t_i$, $\overline{Series1_{ref}}$ is the mean value of the water level time series from the baseline, and $\overline{Series2_{ini}}$ is the mean value of the other water level time series at the same time as $\overline{Series1_{ref}}$.

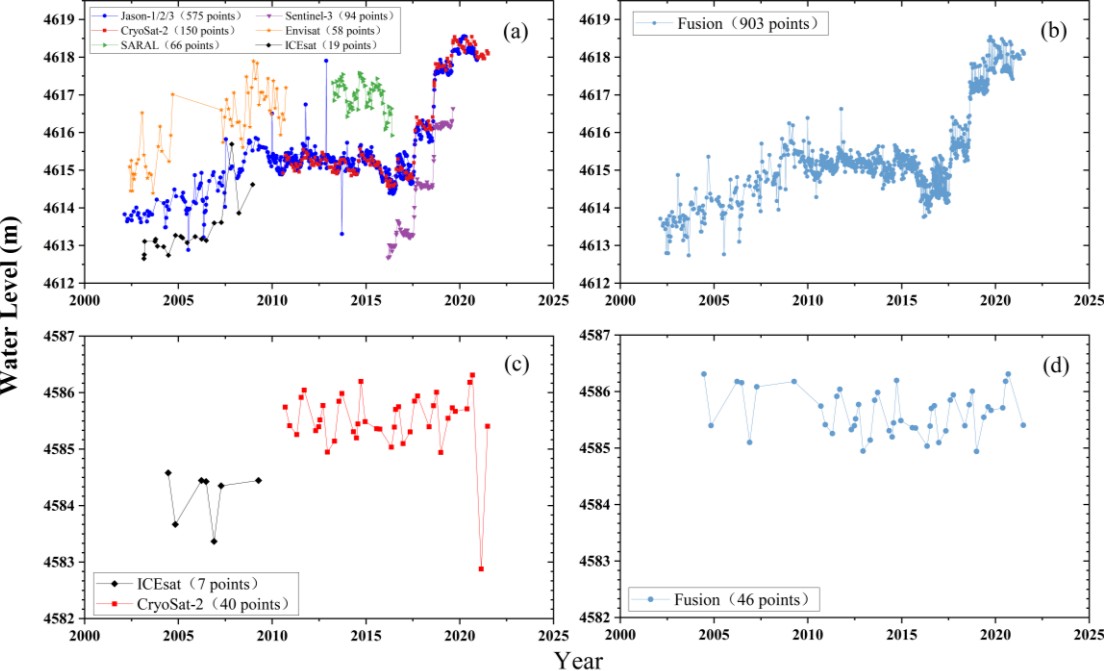

Fig. 4 The process of merging multi-altimetry data. (a) The water level data from eight altimeters in Zhari Namco; (b) The fusion water level data in Zhari Namco ; (c) The water level data from two altimeters in Cuona Lake; (d) The fusion water level data in Cuona Lake.

Nevertheless, not all the lake-level time series can be merged successfully following the steps outlined above. For instance,
Cuona Lake, Xiasa'er Co, and Bei Hulsan Lake cannot be merged successfully because only ICESat and Cryosat-2 were available on these lakes before 2013, while there is no overlap period between ICESat and Cryosat-2. In this study, 18 lakes were found to have similar problems.

A combined linear-periodic-residual model (Liao et al., 2014) was used to simulate and forecast the lake-level time series in the no-overlap period to merge the two altimeters with no overlap period. Numerous studies (Medina et al., 2008; Irvine et al.,
1992; Kropáček et al., 2012; Lee et al., 2011) have indicated that the changes in the lake-level exhibited a clear linear trend and inter-periodic fluctuations at some scales such as 10 or 20 years in line with Eq. (5).

$$x_i = a + bt + \sum_{i=1}^{p} \left( \alpha_i cos \frac{2\pi}{T_i} t + \beta_i sin \frac{2\pi}{T_i} t \right) + \varepsilon_t \tag{5}$$

where $a$ and $b$ are the linear components of the lake-level time series, $T_i$ indicates the $i$th periodic component, and $\varepsilon_t$ is the remaining random component after removal of the linear and periodic components.

A result for the merged altimetry data of Cuona Lake is presented in Fig. 4c and 4d. First, singular spectrum analysis (SSA) algorithms are used to reduce the noise of the lake-level time series and to extract the effective fluctuating signal. Second, we decomposed the fluctuating signal into a linear component, a periodic component, and the remaining residuals using a simple linear fitting, wavelet analysis; then simple regression analysis, trigonometric function fitting, and the autoregressive-moving-average (ARMA) model were used to fit each component, respectively. Finally, we combined the modeling data of each
component and obtain the simulated water level. The diagram for fusion processing is shown in Fig. 5.

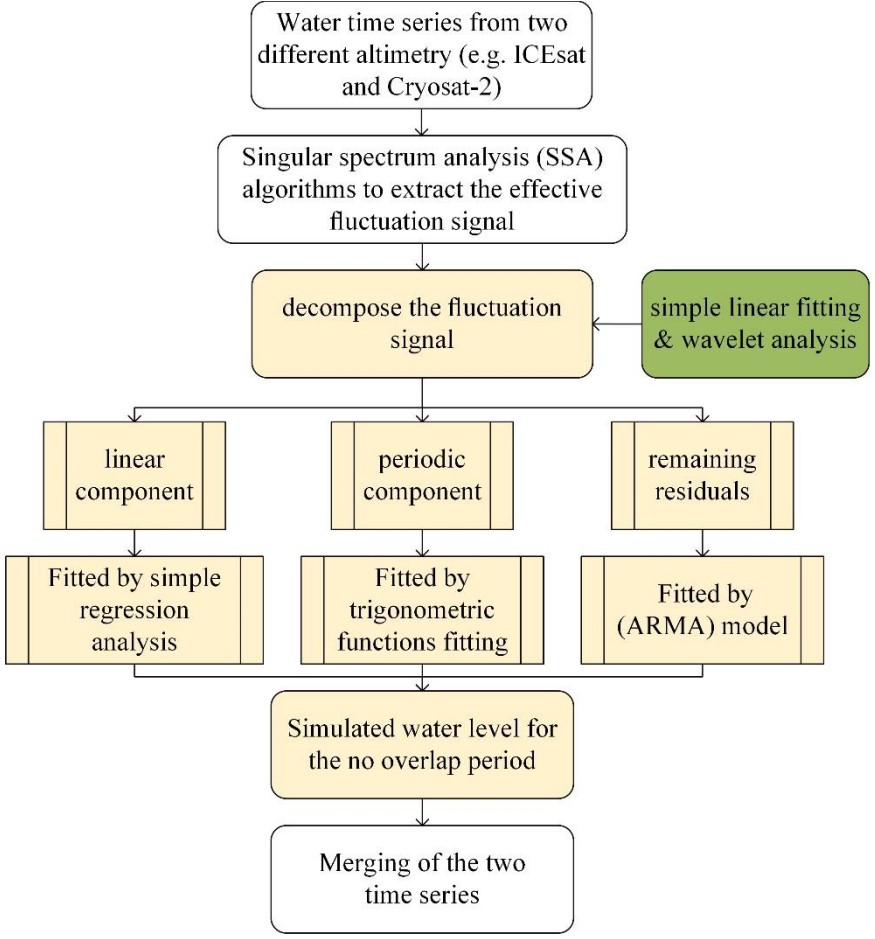

Fig. 5 Flowchart of fusion processing for the water level time series from different altimeters. Steps with a yellow background indicate preparation for merging the time series.

## 4 Validation of data quality

### 4.1 Validation and accuracy of lake level time series

Due to the lack of *in situ* data for the water levels of lakes in the TP, only *in situ* data for eight lakes were collected to validate the accuracy of the lake level time series, and the datums of these *in situ* data were unknown, so the comparison of the water level anomaly between *in situ* data and lake level in this study was performed by removing the mean value over the validation period. Fig. 6 shows the comparison of the water level anomaly between *in situ* data and lake level extracted from altimetry data. It can be seen that there is good consistency between *in situ* data and lake level extracted from altimetry data. Table 4 gives the statistical results for a comparison between the lake level time series and the in-situ data for the eight lakes. The results show that the accuracy for all eight lakes was less than 0.35 m, and the average accuracy was 0.213 m. Dawaco had the lowest root-mean-square errors (RMSEs) (0.149 m), and Ngoring Lake had the highest RMSEs (0.335 m), indicating that the results of this study are reliable and the accuracy of the time series can reach the decimeter level with respect to the monitoring inland lakes. At the same time, except for Dawaco, the lake levels obtained in this study agreed well with those from the *in situ* gauges, showing a good correlation (the correlation coefficients >0.60). Furthermore, it can be seen from the comparison between the satellite-derived lake levels and the *in situ* water levels for the eight lakes that the satellite-derived lake level series followed the gauged data quite well, especially for Qinghai Lake, Bamco, and Pungco (correlation coefficients >0.90).

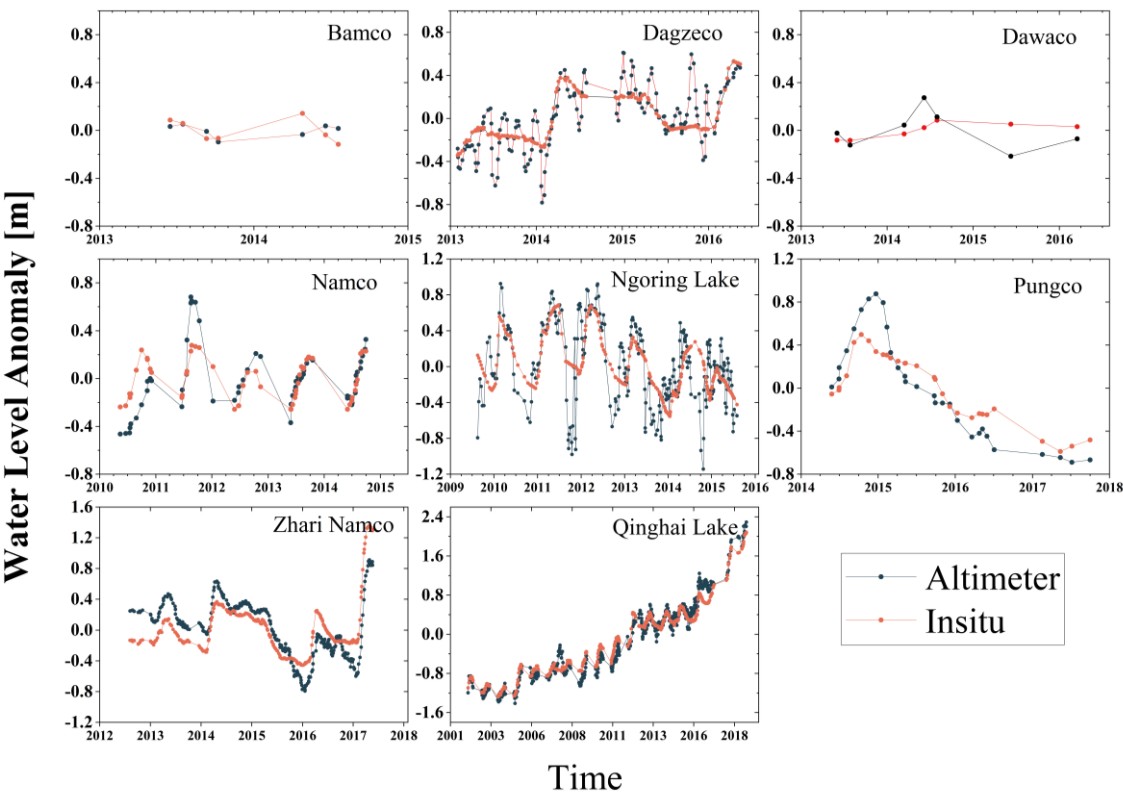

Fig. 6 Comparison of the water level anomaly between *in situ* data and lake level extracted from altimetry data

Table 4 Comparison between the lake levels in this study and the *in situ* water levels

| Lake | Correlation coefficient | RMSE (m) | Number of validation points |
|---|---|---|---|
| Qinghai_Lake | 0.977 | 0.190 | 570 |
| Ngoring_Lake | 0.635 | 0.335 | 284 |
| Bamco | 0.930 | 0.181 | 19 |
| Dagzeco | 0.744 | 0.199 | 156 |
| Dawaco | 0.209 | 0.149 | 7 |
| Namco | 0.738 | 0.179 | 60 |

| | | | |
|---|---|---|---|
| Pungco | 0.924 | 0.222 | 29 |
| Zhari Namco | 0.762 | 0.251 | 314 |

## 4.2 Cross-validation with similar products

We made a comparison of our product with three different lake level datasets provided by DAHITI, LEGOS Hydroweb, and G-REALM (Global Reservoirs and Lakes Monitor). In Figure 7 and Appendix A, we compared the time series of water levels for 46 lakes from DAHITI, 40 lakes from LEGOS Hydroweb, and 8 lakes from G-REALM against the lake levels from our study. The results indicate that the dataset in our study aligns consistently with the other three datasets. The median RMSEs are consistently below 0.30 m (with a value of 0.24 m for DAHITI, a value of 0.27 m for LEGOS Hydroweb, and a value of

0.30 m for G-REALM), while the median correlation values consistently exceed 0.90 (with a value of 0.94 for DAHITI, a value of 0.96 for LEGOS Hydroweb, and a value of 0.96 for G-REALM).

It should be noticed that occasional discrepancies in the statistics may arise from variations in the processing chain for different datasets. For example, Xuelian Lake exhibits an RMSE of 0.79 m when compared to data from DAHITI, whereas it demonstrates a markedly reduced RMSE of 0.29 meters when compared to LEGOS Hydroweb. Moreover, observations for

Zhari Namco across all four datasets reveal that our study's results consistent closely with others, showing an RMSE of approximately 0.30 meters.

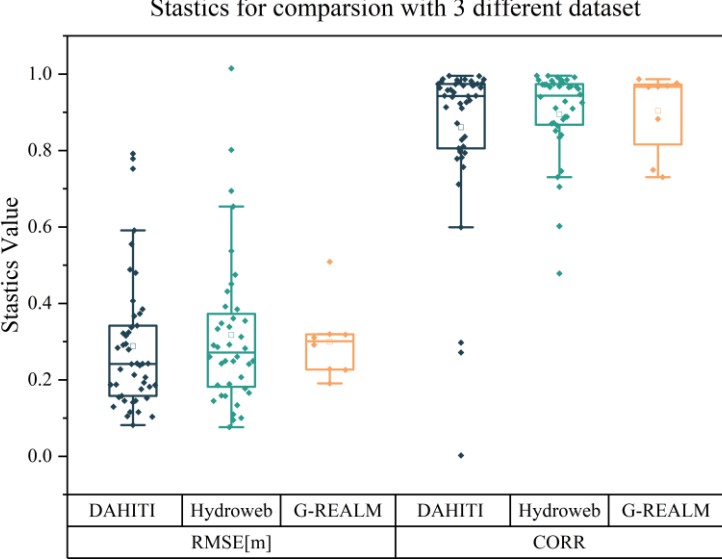

Fig. 7 Cross-validation of the lake levels in the TP derived from the present study with those provided by the DAHITI, LEGOS Hydroweb, and G-REALM.

## 4.3 Description of the data set

The lake-level change time series for 361 lakes (194 lakes for the time series from 2002 to 2021 and 167 lakes for the time series from 2010 to 2021) are available on the datasets. The water level time series for each lake are archived as 361 entities based on the names of the lakes, with a table describing all the information about each lake. The first part of each file describes the basic information of the lake-level time series, such as the geographic information, the start date of the time series, the end

date of the time series, and the number of data points. Next is the main part for each file: the first row stands for the time, the second row records the water level, the third row is the uncertainty of the water level, and the final row stands for the source of the data. It should be noted that the uncertainty of the water level time series was calculated using the standard deviation for the processing in constructing the time series with the "R" package.

**5 Applications**

**5.1 Spatio-temporal analysis of changes in lake levels in the TP**

Based on the changes in the water levels of the lakes (see Appendix B), the spatial patterns for the trends in the lake levels during 2002-2021 are shown in Fig. 8. Overall, the lake levels in the TP show a clear rising trend, and the overall average annual rate of change is 0.175 m/a; further, the number of lakes with rising water levels accounts for 78% of all lakes. The total area of lakes with rising water levels (35213 km$^2$) is much larger than the total area of lakes with falling water levels (6364km$^2$), indicating that the water storage of lakes on the TP is growing. From the distribution of the annual average rate of change of lake levels (Fig. 9), among the monitored lakes between 2002 and 2021, there are more lakes with rising water levels than those with falling water levels. Among the lakes with an average annual rate of change greater than 0.20 m/a, the number of lakes with an increasing trend in the water levels is much higher (280 lakes) than the number of lakes with a decreasing trend (81 lakes).

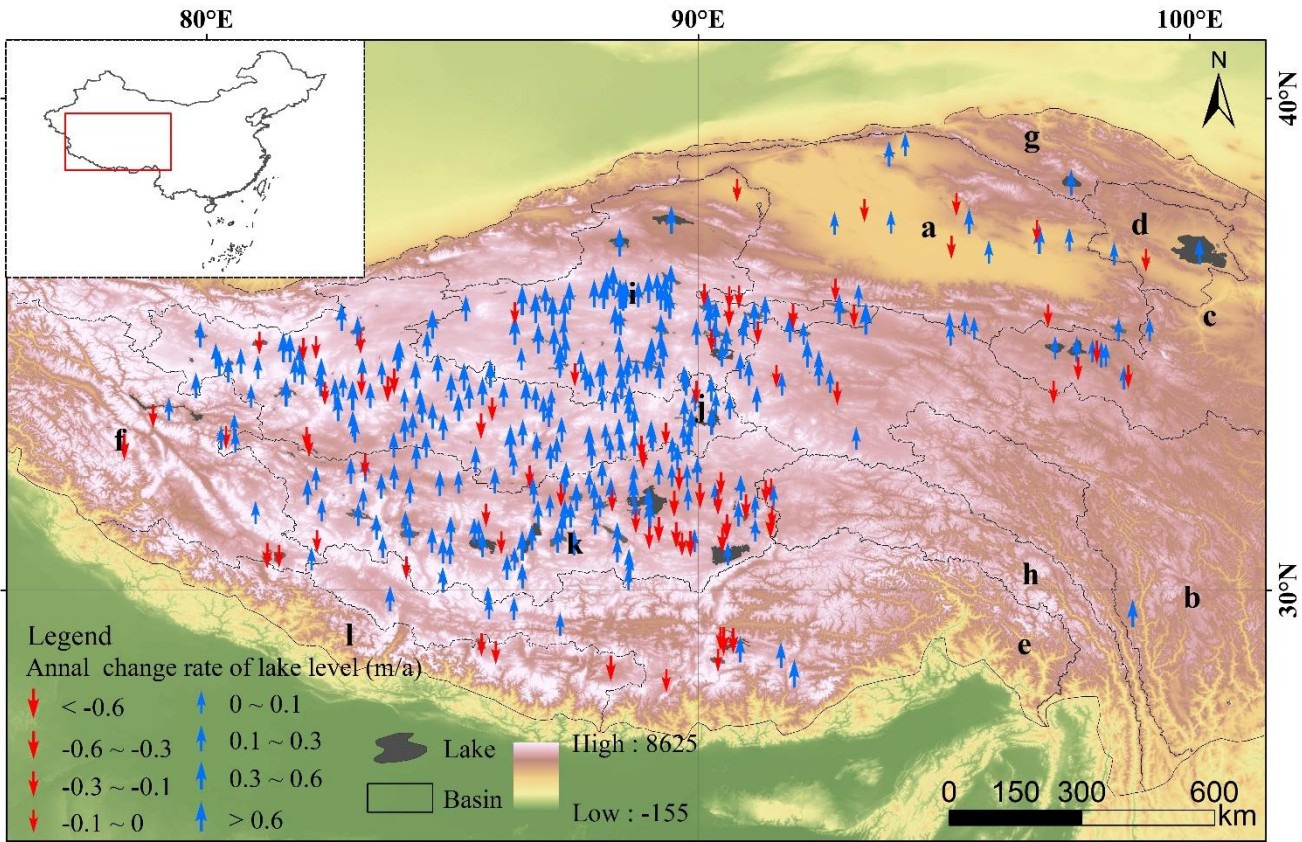

Fig. 8 Spatial distribution of trends in the changes in the water levels of lakes on the TP during 2002-2021. The black line shows the boundary of the basin of the TP (refered to Wan et al., 2016). The lowercase letters indicate different basins. The DEM of the base map is from the Global Multi-resolution Terrain Elevation Data 2010(GMTED2010)(GMTED: https://topotools.cr.usgs.gov/gtmed_viewer/)

(a Qaidam; b Yangtze River; c Yellow River; d Qinghai Lake; e Brahmaputra River; f. Indus River; g Hexi Corridor; h Nu Jiang River; i Northern Inner Plateau; j Central Inner Plateau; k Southern Inner Plateau; l Ganges River)

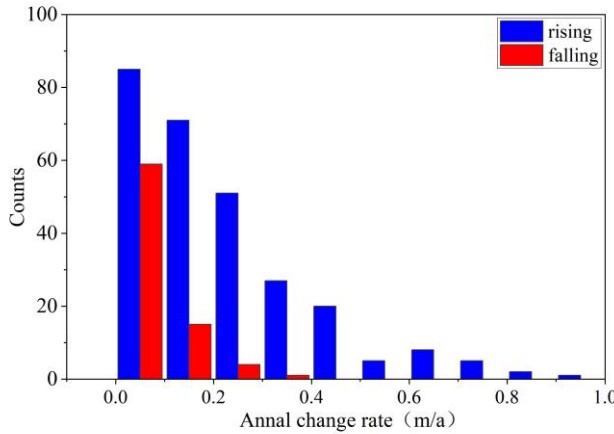

Fig. 9 Histogram of trends in the lake level changes on the TP during 2002-2021.

Analysis of the trends in the changes in the water levels based on the lake areas shows that there is a clear rising trend in the water level of lakes on the TP, the most significant trends in the case of rising water levels being for larger-size lakes (>500 km$^2$) and also for smaller size (<50 km$^2$) lakes, and intermediate size lakes showing significant rising trend (Table 5).

Table 5 The trends for changes in the lake water levels in the TP during 2002-2021

| Lake area/km$^2$ | No. of lakes | Annual rate of change (m/a) | No. of lakes with rising water levels | Mean rate of rise (m/a) | No. of lakes with decreasing water levels | Mean rate of decrease (m/a) |
|---|---|---|---|---|---|---|
| >500 | 13 | 0.167 | 12 | 0.198 | 1 | -0.201 |
| [200,500] | 31 | 0.186 | 24 | 0.262 | 7 | -0.075 |
| [100,200) | 48 | 0.232 | 36 | 0.333 | 12 | -0.069 |
| [50,100) | 69 | 0.157 | 50 | 0.243 | 19 | -0.069 |
| [10,50) | 200 | 0.142 | 159 | 0.195 | 42 | -0.058 |

To better understand the spatial distribution pattern of the changes in the water levels of the lakes, the trends for the changes in the water levels of the lakes in each basin of the TP were analyzed (Table 6). Overall, during the period 2002-2021, the water levels of the lakes in all basins increased significantly, except for the Brahmaputra River Basin. The area of lakes with

rising water levels was larger than that for lakes with decreasing water levels. Amongst all the basins, the lakes with a decreasing water level were distributed mainly in the Brahmaputra River, Ganges River, and Nujiang River Basins (Fig. 10). The water level changes in lake for each basin can be summarized as follows:

*Qaidam Basin*. A total of 22 lakes were monitored in the basin, of which 13 lakes showed a rising trend, with an average rising rate of 0.115m/a and a total rising lake area of 986km$^2$. The other 9 lakes showed a falling trend, with an average falling rate

of -0.027m/a and a total falling lake area of 685km$^2$. The fastest rising lake in the basin is Tuosu Lake with an average annual rate of 0.724m/a and the fastest declining lake is S63005 with an average annual rate of -0.087m/a. The largest lake monitored in the basin is Dabsan Lake with an average annual rate of -0.053m/a.

*Yangtze River Basin*. 15 lakes were measured in the basin. Among these lakes, 12 lakes showed a rising trend with a mean rate of 0.158m/a and a total rising lake area of 590km$^2$. The remaining 3 lakes showed a declining trend with an average rate of -

0.008m/a and the total falling area of 97km$^2$. In the basin, Mang Co has the fastest rising water level with a mean rate of

0.461m/a, and Mazhangcuoqin has the fastest decline trend of -0.021m/a. Yelusu Lake is the largest lake in the basin with an average annual rate of 0.034m/a.

*Qinghai Lake Basin*. 3 lakes were measured in the basin, of which 2 lakes showed the rising water level with a mean rising rate of 0.124m/a. Caka Salt Lake has the decreasing water level with a mean rate of -0.005m/a. Qinghai Lake is the largest lake in this basin with the fastest rising trend of 0.190m/a.

*Yellow River Basin*. The water levels of 11 lakes were monitored in the basin, of which 7 lakes showed a rising trend with an average rate of 0.069m/a, and the other 4 lakes showed a decreasing trend with an average rate of -0.019m/a. In this basin, Ayongwuerma Co has the fastest rising water level with a mean rate of 0.174/a, and Xinxin Lake has the fastest declining water level with a mean rate of -0.053m/a. The largest lake is Kuhai Lake with a mean rate of 0.099m/a.

*Brahmaputra River Basin*. A total of 13 lakes were monitored in the basin, mainly in the upper and middle reaches of the Brahmaputra River. The water levels of 7 lakes showed a rising trend with an average rising rate of 0.163m/a, and the water levels of the remaining 6 lakes showed a falling trend with an average falling rate of -0.114m/a. In this basin, Nariyong Co has the fastest rising water level with a rising rate of 0.441m/a, and Chen Co has the fastest falling water level with a falling rate of -0.349m/a. Yamzho Yumco is measured the largest lake in the basin, and it has an average falling rate of -0.201m/a.

*Indus River Basin*. 8 lakes were measured in this basin, mainly distributed in the northeastern and northwestern basin. Among them, 4 lakes had a rising water level with a rising rate of 0.062m/a, and the other 4 lakes had a decreasing water level with a falling rate of -0.077m/a. In this basin, Bangong Co has the fastest rising water level with an average rate of 0.092m/a, and Langa Co has the fastest falling water level with an average rate of -0.156m/a. Mapam Yumco is measured the largest lake with an average falling rate of -0.013m/a.

*Inner Plateau Basin*. The basin contains the Qiangtang Plateau and the Cocosili region, with a harsh natural environment and dry climate, and is the largest endorheic area on the TP. The water levels of 282 lakes were monitored in the basin, and 233 lakes have a rising trend with an average rising rate of 0.249m/a. The remaining 49 lakes have a declining trend, mainly in the southeast and northwest areas of the basin, with an average falling rate of -0.074m/a. The fastest rising lake in the basin is Yan Lake with an average rate of 2.384m/a, and the fastest falling lake is Dongka Co with an average rate of -0.266m/a. Seling Co is measured the largest lake, and its average annual rate of change in water level is 0.304m/a.

In addition, since the number of lakes monitored in the Nujiang River, Ganges River and Hexi Corridor Basins is very small, their analysis have not be conducted.

Table 6 The trends in the changes in the water levels of the lakes in the different basins of the TP during 2002-2021

| Basin | No. of lakes | No. of lakes with rising water levels | Annual rate of rise (m/a) | Area of lakes with rising water levels (km$^2$) | No. of lakes with decreasing water levels | Annual rate of fall (m/a) | Area of lakes with decreasing water levels (km$^2$) |
|---|---|---|---|---|---|---|---|
| Qaidam | 22 | 13 | 0.115 | 986 | 9 | -0.027 | 685 |
| Yangtze River | 15 | 12 | 0.158 | 590 | 3 | -0.008 | 97 |
| Yellow River | 11 | 7 | 0.069 | 1285 | 4 | -0.019 | 82 |
| Qinghai Lake | 3 | 2 | 0.124 | 4391 | 1 | -0.005 | 115 |
| Brahmaputra River | 13 | 7 | 0.163 | 224 | 6 | -0.114 | 1048 |
| Indus River | 8 | 4 | 0.062 | 762 | 4 | -0.077 | 869 |

| | | | | | | | |
|---|---|---|---|---|---|---|---|
| Northern Inner Plateau | 80 | 73 | 0.378 | 7285 | 7 | -0.079 | 506 |
| Central Inner Plateau | 104 | 85 | 0.236 | 5681 | 19 | -0.050 | 1013 |
| Southern Inner Plateau | 98 | 75 | 0.137 | 13617 | 23 | -0.094 | 1466 |
| Nujiang River | 3 | 1 | 0.003 | 17 | 2 | -0.008 | 212 |
| Ganges River | 3 | 0 | / | / | 3 | -0.101 | 335 |
| Hexi Corridor | 1 | 1 | 0.189 | 609 | 0 | / | / |

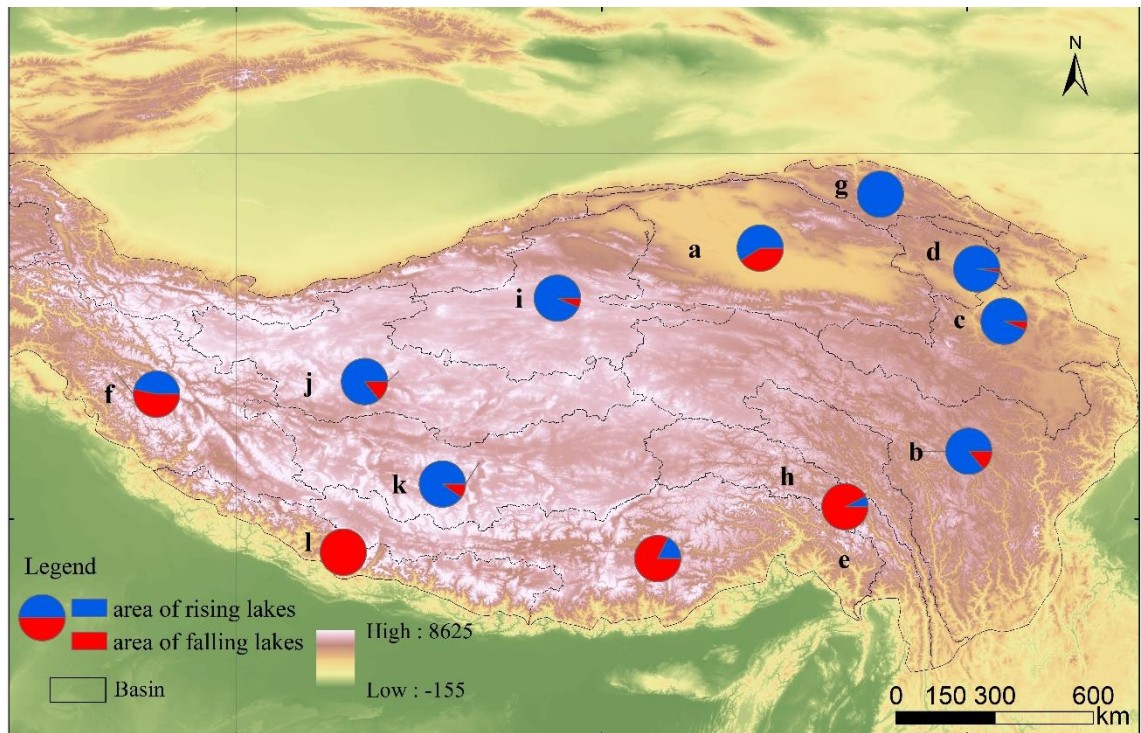

Fig. 10 Relative proportions of the trends in the changing levels of water in the lakes in each basin. The boundry of each basin is referred to Wan et al.(2016). The DEM of the base map is from the Global Multi-resolution Terrain Elevation Data 2010(GMTED2010)(GMTED: https://topotools.cr.usgs.gov/gtmed_viewer/) (the lowercase letters indicate the different

lake basins studied as in Fig. 8).

**5.2 Exploring the responses of the lake levels to river regulation**

Aided by the availability of the high-space-coverage lake level time series, it is possible to explore the responses of the water levels in the lakes to river regulation to provide support for integrated management of the lake water resources. Here, the streamflow of the rivers and lakes in the source area of the Yellow River are taken as an example to analyze the relationships

between changes in the streamflow of the source area of the Yellow River and the Nogring Lake and Gyaring Lake. From inspection of Fig. 11 the discharge at the source of the Yellow River is directly affected by the regulation and storage of these two lakes, which results in changes to the annual distribution of the discharge at the source; thus, the discharge along the Yellow River is more uniform than that in the lower reaches of Tangnaihai and Xunhua (Fig. 12). However, the correlation between precipitation and the changes in the discharge at the source of the Yellow River is very poor, and there is a certain

time lag. The changes in water levels of the lakes are basically synchronized with the changes in the streamflow at the source of the Yellow River (Fig. 12), indicating that the streamflow is interannual regulated.

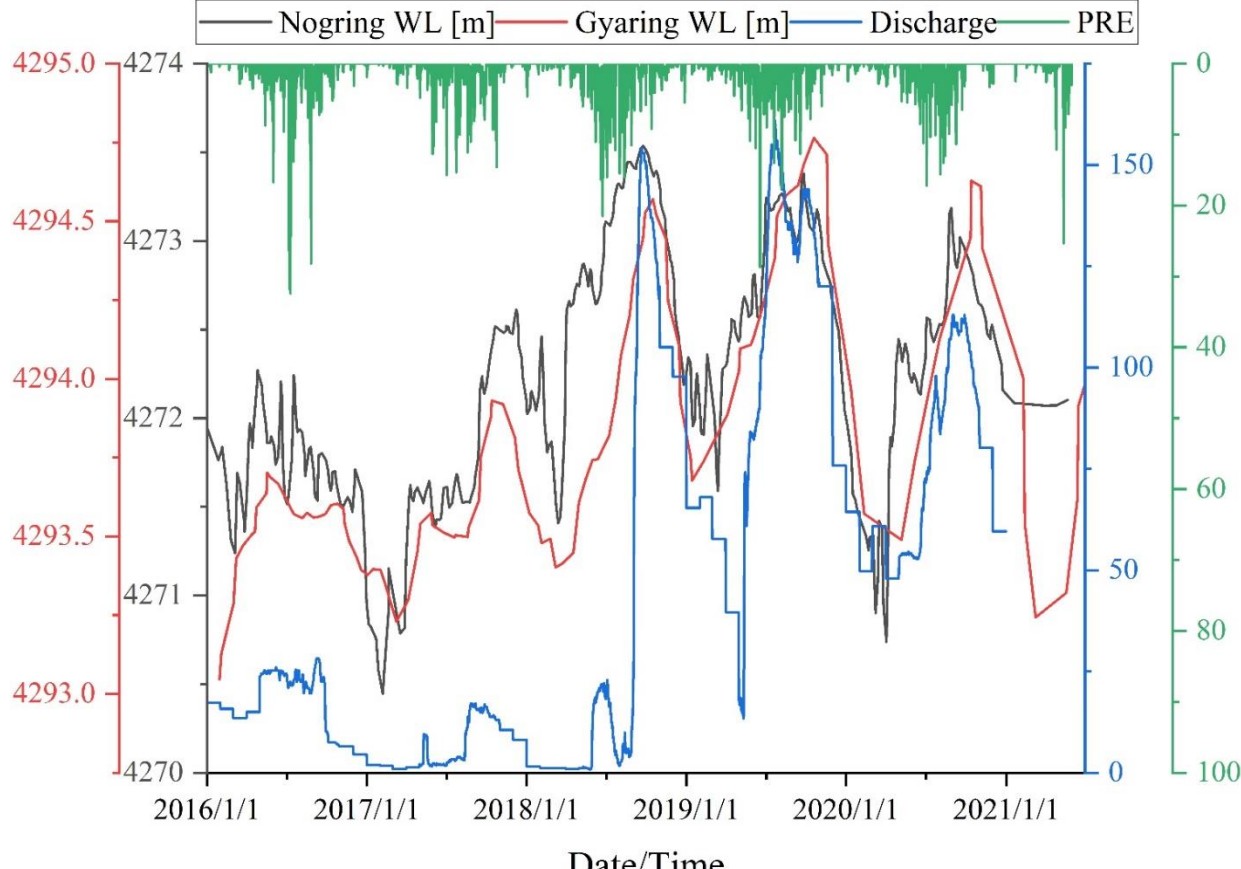

Fig. 11 Responses of water levels in the lakes to regulation of streamflow in the river (Nogring WL represents the water level of Nogring Lake; Gyaring WL represents the water level of Gyaring Lake; PRE represents precipitation).

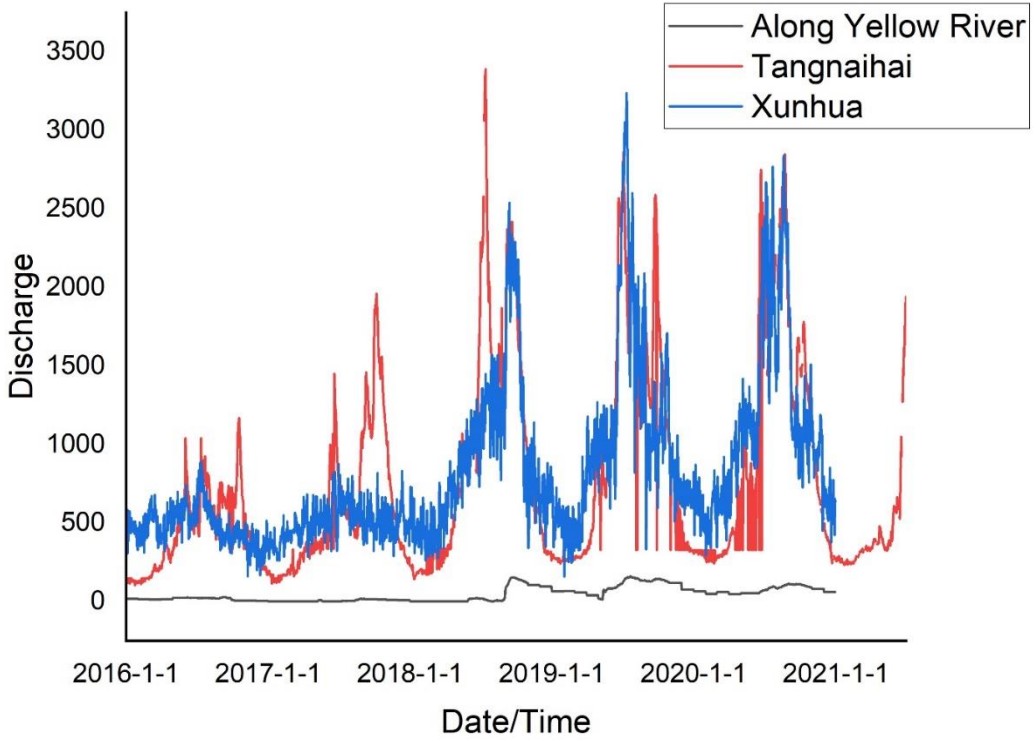

Fig. 12 Changes in the discharge along the Yellow River and the lower reaches of Tangnaihai and Xunhua.

**6 Data availability**

The derived water levels in the lakes of the TP are archived and available at https://doi.org/10.1594/PANGAEA.939427 (Chen et al., 2021).

**7 Conclusion**

In this study, high-resolution datasets for changes in the water levels for 361 lakes on the TP during 2002-2021 were developed based on multi-altimeter data from Envisat, ICESat-1, CryoSat-2, Jason-1, Jason-2, Jason-3, SARAL, and Sentinel-3A. A modified waveform retracker and a noise-footprint removal method were used to extract the water levels, and the lake level time series were then estimated using the R package tsHydro. The dynamic reference time series was then used to merge the lake-level time series from the multi-altimeter data. It was found that the merged water levels based on the altimetry increased the overall sampling frequency regardless of the lake size. The water levels derived from the altimeter data were validated with *in situ* data, and the accuracy of the time series for monitoring lakes reached the decimeter level. Based on comparison with the DAHITI, LEGOS Hydroweb, and G-REALM datasets, the new product was found to be consistent with these products, and the median RMSEs are consistently below 0.30 m, while the median correlation values consistently exceed 0.90, indicating that the new dataset was reliable. In addition, the spatio-temporal changes in the water levels of the lakes on the TP during 2002-2021 were explored. Overall, the measured lake levels on the TP were indicative of a rising trend with an overall average annual rate of change of 0.175 m/a; moreover, the number of lakes with rising water levels accounted for 78% of the total examined. The lakes with the most significant rises in the water levels were those of large size (>500 km$^2$) and small size (<50 km$^2$), and the intermediate size lakes showed the significant rising trend in the water levels. The water levels of lakes in all basins have been increasing significantly over the period 2002 to 2021 except for the Brahmaputra River Basin. The lakes with decreasing water levels were distributed mainly in Brahmaputra River, Ganges River, and Nujiang River Basins. Further applications of the lake level dataset of the TP are anticipated. For example, the dataset may be used to analyze the responses of the lake levels to river regulation to provide support for managing lake water resources.

**Author contributions**

Liao J and Chen J designed the research plan. Chen J developed the approaches and the dataset. Liao J, Lou Y and Ma S contributed to the analysis of the results. Shen G and Zhang L contributed to the data processing. Liao J and Chen J wrote the manuscript.

**Competing interests.** The authors declare that there are no conflicts of interest.

**Acknowledgements**

We thank the European Space Agency and Centre National d'Etudes Spatiales for providing the altimeter data and the Bureau of Hydrology and Water Resources of Qinghai Province, the Yellow River Commission of the Ministry of Water Resources, and the Institute of Tibetan Plateau Research, Chinese Academy of Sciences for providing *in situ* gauge measurements of the water levels.

**Financial support**

This work was supported by the National Natural Science Foundation of China (Grant 41871256).

**Review statement**

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

Appendix A. Comparison of the lake levels in the TP derived from this study with those provided by the DAHITI, LEGOS Hydroweb, and G-REALM in RMSE and Correlation.

| Lake Name | DAHITI ID | RMSE | CORR | NP* | Lake Name | DAHITI ID | RMSE | CORR | NP |
|-----------|-----------|------|------|-----|-----------|-----------|------|------|-----|
| Ake Sayi Lake | 10445 | 0.48 | 0.94 | 83 | Luotuo Lake | 10538 | 0.32 | 0.84 | 44 |
| Aqqujjik Kaje | 11004 | 0.08 | 0.99 | 44 | Ma'erxia Co | 10986 | 0.21 | 0.92 | 49 |
| Ayakkum Lake | 10540 | 0.37 | 0.99 | 123 | Meiriqiecuomari | 10556 | 0.24 | 0.93 | 49 |
| Bairab Co | 11036 | 0.13 | 0.81 | 49 | Mugqu Co | 11018 | 0.19 | 0 | 5 |
| Chabo Co | 10543 | 0.14 | 0.91 | 36 | Nam Co | 345 | 0.15 | 0.94 | 110 |
| Chibzhang Co | 41056 | 0.16 | 0.8 | 11 | Ngangla Ringco | 10537 | 0.15 | 0.98 | 53 |
| Dagze Co | 10425 | 0.78 | 0.97 | 149 | Ngangze Co | 10404 | 0.28 | 0.96 | 387 |
| Dangqiong Co | 11019 | 0.37 | 0.78 | 73 | Orba Co | 11477 | 0.24 | 0.87 | 116 |

| Lake Name | | RMSE | CORR | NP | Lake Name | | RMSE | CORR | NP |
|---|---|---|---|---|---|---|---|---|---|
| Daxiong Lake | 11053 | 0.12 | 0.98 | 41 | Pung Co | 10975 | 0.41 | 0.97 | 83 |
| Deyu Lake | 11015 | 0.21 | 0.96 | 49 | Qiagui Co | 10989 | 0.49 | 0.3 | 72 |
| Dulishi Lake | 11126 | 0.1 | 0.98 | 50 | Qinghai Lake | 227 | 0.19 | 0.99 | 366 |
| Garen Co | 11030 | 0.24 | 0.83 | 38 | Selin Co | 233 | 0.19 | 1 | 153 |
| Garkung Caka | 11001 | 0.1 | 0.98 | 47 | Serbug Co | 11073 | 0.34 | 0.6 | 19 |
| Goren Co | 10536 | 0.34 | 0.8 | 79 | Sugan Lake | 11005 | 0.28 | 0.27 | 29 |
| Gozha Co | 10448 | 0.19 | 0.81 | 46 | Tangra Yumco | 10424 | 0.29 | 0.97 | 203 |
| Har Lake | 10419 | 0.23 | 0.98 | 155 | Taro Co | 10421 | 0.24 | 0.96 | 22 |
| Heishi North Lake | 11070 | 0.32 | 0.79 | 16 | Tu Co | 10973 | 0.15 | 0.99 | 87 |
| Jieze Caka | 10427 | 0.12 | 0.93 | 43 | Wanquan Lake | 11037 | 0.59 | 0.78 | 125 |
| Jingyu Lake | 10995 | 0.38 | 0.96 | 33 | Xiangyang Lake | 11012 | 0.55 | 0.97 | 48 |
| Kyebxang Co | 11025 | 0.24 | 0.76 | 37 | Xuelian Lake | 11040 | 0.79 | 0.71 | 92 |
| Lagkor Co | 11020 | 0.18 | 0.94 | 48 | Xuru Co | 10105 | 0.18 | 0.94 | 45 |
| Longwei Co | 11003 | 0.15 | 0.97 | 46 | Yaggain Co2 | 11035 | 0.32 | 0.91 | 40 |
| Lumajiangdong Co | 10426 | 0.75 | 0.95 | 66 | Zhari Namco | 10423 | 0.29 | 0.97 | 443 |

| Lake Name | Legos ID | RMSE | CORR | NP | Lake Name | Legos ID | RMSE | CORR | NP |
|---|---|---|---|---|---|---|---|---|---|
| Ake Sayi Lake | 1300000001373 | 0.36 | 0.97 | 50 | Lumajiangdong Co | 1300000001399 | 1.01 | 0.84 | 71 |
| Aqqujjik Kaje | 1300000001352 | 0.09 | 0.99 | 22 | Luotuo Lake | 1300000014972 | 0.31 | 0.85 | 43 |
| Ayakkum Lake | 1300000001344 | 0.34 | 0.99 | 151 | Mapam Yumco | 1300000001454 | 0.35 | 0.73 | 56 |
| Bairab Co | 1300000001379 | 0.11 | 0.83 | 45 | Nam Co | 1300000000149 | 0.21 | 0.88 | 201 |
| Bangong Co | 1300000001403 | 0.25 | 0.60 | 72 | Ngangla Ringco | 1300000001431 | 0.35 | 0.48 | 88 |
| Chabo Co | 1300000015037 | 0.19 | 0.71 | 11 | Ngangze Co | 1300000001447 | 0.24 | 0.97 | 328 |
| Chibzhang Co | 1300000001404 | 0.25 | 0.99 | 291 | Ngoring Lake | 1300000001377 | 0.45 | 0.87 | 199 |
| Cuoda Rima | 1300000014898 | 0.29 | 0.91 | 46 | Orba Co | 1300000014959 | 0.33 | 0.75 | 73 |
| Dagze Co | 1300000001425 | 0.65 | 0.98 | 147 | Pung Co | 1300000001433 | 0.69 | 0.89 | 44 |
| Dangqiong Co | 1300000015180 | 0.13 | 0.97 | 29 | Qinghai Lake | 1300000000143 | 0.18 | 0.97 | 204 |
| Dogai Coring | 1300000001389 | 0.38 | 0.93 | 111 | Selin Co | 1300000000147 | 0.17 | 1.00 | 265 |
| Dogaicoring Qangco | 1300000001372 | 0.39 | 0.98 | 96 | Tangra Yumco | 1300000001450 | 0.28 | 0.97 | 205 |
| Garkung Caka | 1300000015010 | 0.10 | 0.97 | 36 | Taro Co | 1300000001445 | 0.29 | 0.87 | 34 |
| Goren Co | 1300000001439 | 0.26 | 0.86 | 21 | Telashi Lake | 1300000014940 | 0.24 | 0.94 | 50 |
| Hoh Xil Lake | 1300000001369 | 0.16 | 1.00 | 16 | Tu Co | 1300000001405 | 0.16 | 0.98 | 35 |
| Huolunuo'er | 1300000001370 | 0.14 | 0.93 | 41 | Urru Co | 1300000001428 | 0.47 | 0.60 | 47 |
| Jieze Caka | 1300000001401 | 0.08 | 0.97 | 32 | Wulanwula Lake | 1300000001386 | 0.43 | 0.96 | 126 |
| Jingyu Lake | 1300000001357 | 0.54 | 0.97 | 28 | Xuelian Lake | 1300000015002 | 0.29 | 0.89 | 43 |
| Langa Co | 1300000001452 | 0.19 | 0.91 | 310 | Zhari Namco | 1300000001449 | 0.25 | 0.98 | 496 |
| Lexiewudan Co | 1300000001366 | 0.80 | 0.97 | 109 | Zige Tangco | 1300000001422 | 0.26 | 0.95 | 202 |

| Lake Name | G-REALM ID | RMSE | CORR | NP | Lake Name | G-REALM ID | RMSE | CORR | NP |
|---|---|---|---|---|---|---|---|---|---|
| Bangong Co | lake000121 | 0.23 | 0.73 | 341 | Chibzhang Co | lake000171 | 0.23 | 0.99 | 448 |
| Langa Co | lake000141 | 0.29 | 0.97 | 533 | Orba Co | lake000177 | 0.32 | 0.75 | 290 |
| Zhari Namco | lake000152 | 0.32 | 0.97 | 483 | Dogai Coring | lake000189 | 0.19 | 0.98 | 389 |
| Ngangze Co | lake000156 | 0.31 | 0.97 | 568 | Ngoring Lake | lake000285 | 0.51 | 0.88 | 452 |

*NP indicates number of points for validation

515

Appendix B, Supplementary data

| No. | Lake Name | Lat. (deg) | Lon. (deg) | Area (km$^2$) | Duration (yyyy/mm/dd) | Annual rate (m/a) | P-value | Altimeter type* |
|---|---|---|---|---|---|---|---|---|
| 1 | Ake Sayi Lake | 35.2 | 79.86 | 258.25 | 2003/04/20-2021/07/17 | 0.1837 | < 0.001 | 1,2,3,7,8 |
| 2 | Amu Co | 33.49 | 88.7 | 114.98 | 2007/03/23-2021/05/11 | 0.2746 | < 0.001 | 1,3,5,7 |
| 3 | Angrenjin Co | 29.31 | 87.19 | 21.08 | 2016/04/29-2021/01/12 | 0.0540 | < 0.001 | 3,8 |
| 4 | Angshang Co | 33.72 | 82.67 | 27.66 | 2007/10/13-2021/05/23 | 0.3547 | < 0.001 | 2,3,8 |
| 5 | Aqqujjik Kaje | 37.07 | 88.4 | 350 | 2003/10/13-2021/07/27 | 0.5355 | < 0.001 | 1,2,3,7,8 |
| 6 | Argog Co | 30.98 | 82.24 | 55.26 | 2003/09/16-2020/08/28 | -0.0104 | 0.400 | 1,2,3 |
| 7 | Aru Co | 33.99 | 82.4 | 104.32 | 2003/10/06-2021/06/20 | -0.0198 | < 0.001 | 1,2,3,7 |
| 8 | Ayakkum Lake | 37.53 | 89.45 | 520 | 2003/01/02-2021/07/25 | 0.3262 | < 0.001 | 1,2,3,4,5,7,8 |
| 9 | Bangdag Co | 34.94 | 81.56 | 142.92 | 2005/06/17-2021/05/28 | 0.6624 | < 0.001 | 2,3,7 |
| 10 | Bangkog Co | 31.74 | 89.51 | 123.87 | 2003/03/11-2021/06/29 | -0.1595 | < 0.001 | 1,3,7,8 |
| 11 | Bangong Co | 33.68 | 79.23 | 671.2 | 2002/10/26-2021/06/27 | 0.0919 | < 0.001 | 1,2,3,4,5,6,7,8 |
| 12 | Bensong Co | 33.21 | 86.43 | 15.27 | 2007/04/13-2016/03/14 | 0.2540 | 0.016 | 1,7 |
| 13 | Bong Co | 31.22 | 91.16 | 143.98 | 2011/03/28-2021/06/02 | 0.0153 | 0.222 | 1,3,7 |
| 14 | Buergacuo Lake | 33.66 | 84.38 | 10.01 | 2003/09/13-2019/11/22 | 0.2335 | < 0.001 | 1,3,5,7 |
| 15 | Cam Co | 32.12 | 83.55 | 103.7 | 2009/08/27-2020/08/26 | 0.2116 | < 0.001 | 3,4,5 |
| 16 | Cedo Caka | 33.17 | 89.04 | 74.96 | 2008/02/22-2021/07/02 | 0.3690 | < 0.001 | 1,2,3,8 |
| 17 | Cemar Co | 33.55 | 84.59 | 49.42 | 2012/04/13-2020/09/19 | 0.1542 | < 0.001 | 3 |
| 18 | Chabo Co | 33.36 | 84.19 | 49.47 | 2007/10/29-2020/06/06 | 0.1417 | < 0.001 | 1,2,3,5,8 |
| 19 | Changhu Lake1 | 35.02 | 84.48 | 10.35 | 2007/04/08-2021/06/17 | 0.1169 | < 0.001 | 2,3 |
| 20 | Changhu Lake2 | 34.71 | 89.04 | 51.08 | 2003/12/18-2021/07/25 | 0.1420 | < 0.001 | 1,3,7 |
| 21 | Chaxiabucuo Lake | 31.93 | 87.88 | 11.53 | 2007/10/24-2021/05/13 | 0.1452 | < 0.001 | 2,3 |
| 22 | Chem Co | 34.16 | 79.78 | 121.53 | 2007/03/24-2021/05/05 | 0.1460 | < 0.001 | 2,3,5 |
| 23 | Chibzhang Co | 33.45 | 90.27 | 541.18 | 2003/03/03-2021/07/22 | 0.4185 | < 0.001 | 1,2,3,4,5,6,7,8 |
| 24 | Co Ngoin1 | 31.59 | 88.72 | 268.42 | 2007/11/02-2021/07/25 | 0.0135 | 0.090 | 1,2,3,8 |
| 25 | Co Nyi | 34.55 | 87.18 | 166.91 | 2005/06/15-2021/07/30 | 0.0988 | < 0.001 | 1,2,3,7 |

| 26 | Cuoda Rima | 35.33 | 91.86 | 83.87 | 2005/03/21-2021/03/16 | 0.3154 | < 0.001 | 3,4,5,6 |
|----|-----------|-------|-------|-------|-----------------------|--------|---------|---------|
| 27 | Cuona Co | 31.63 | 82.34 | 52.81 | 2007/03/23-2021/04/05 | 0.0374 | 0.066 | 2,3 |
| 28 | Cuona Lake | 32.03 | 91.48 | 191.46 | 2011/07/17-2021/06/25 | -0.0117 | 0.051 | 3 |
| 29 | Dabsan Lake | 36.96 | 95.15 | 296.4 | 2009/08/06-2021/05/23 | -0.0530 | < 0.001 | 1,3,4,5,7,8 |
| 30 | Daggyai Co | 29.84 | 85.72 | 109.43 | 2005/11/08-2021/07/07 | 0.0622 | 0.016 | 1,2,3 |
| 31 | Dagze Co | 31.89 | 87.52 | 311.04 | 2003/02/24-2021/07/02 | 0.4180 | < 0.001 | 1,2,3,5,6,7,8 |
| 32 | Damazirang | 30.95 | 85.99 | 32.98 | 2011/12/15-2021/06/12 | -0.0124 | 0.323 | 1,3 |
| 33 | Dangqiong Co | 31.57 | 86.74 | 63.87 | 2010/01/04-2019/08/04 | 0.1480 | < 0.001 | 1,7,8 |
| 34 | Dangquezangbu | 29.83 | 83.73 | 62.6 | 2005/02/26-2021/07/12 | 0.1130 | < 0.001 | 1,2,3,7,8,37 |
| 35 | Darab Co | 32.47 | 83.22 | 25.66 | 2005/10/29-2020/10/15 | 0.1259 | < 0.001 | 2,3 |
| 36 | Dawa Co | 31.24 | 84.96 | 118.2 | 2007/04/05-2021/06/15 | 0.2620 | < 0.001 | 2,3 |
| 37 | Daxiong Lake | 34.05 | 85.61 | 42.93 | 2008/10/14-2021/05/20 | 0.3077 | < 0.001 | 2,3,8 |
| 38 | Deyu Lake | 35.69 | 87.27 | 61.63 | 2004/05/28-2021/07/30 | 0.3648 | < 0.001 | 1,2,3,4,5,7,8 |
| 39 | Dogai Coring | 34.58 | 88.96 | 492.4 | 2002/11/28-2021/07/23 | 0.2257 | < 0.001 | 1,2,3,4,5,6,7,8 |
| 40 | Dogaicoring Qangco | 35.32 | 89.24 | 403.1 | 2003/03/14-2021/07/25 | 0.3900 | < 0.001 | 1,3,7,8 |
| 41 | Dong Co | 32.18 | 84.74 | 105.43 | 2004/01/12-2021/04/03 | 0.1468 | < 0.001 | 1,3,7 |
| 42 | Donggei Cuona Lake | 35.3 | 98.55 | 241.37 | 2003/02/04-2021/06/14 | 0.0651 | < 0.001 | 1,3,7,8 |
| 43 | Dulishi Lake | 34.73 | 81.89 | 98.55 | 2003/11/28-2021/04/05 | 0.2853 | < 0.001 | 1,3,7,8 |
| 44 | Dung Co | 31.71 | 91.16 | 151.44 | 2010/05/26-2021/06/02 | 0.0020 | 0.750 | 1,3,5,6,8 |
| 45 | Duoqing Co | 28.15 | 89.35 | 49.6 | 2003/07/10-2021/05/11 | -0.0271 | 0.010 | 1,3,7,8 |
| 46 | Finger Lake | 33.72 | 85.12 | 15.18 | 2004/04/26-2021/07/10 | 0.3339 | < 0.001 | 1,3,7 |
| 47 | Gangnagema Co | 34.32 | 98.66 | 32.03 | 2012/06/06-2020/07/01 | 0.0136 | 0.109 | 3 |
| 48 | Gansenquan Lake | 37.46 | 92.77 | 20.02 | 2008/03/08-2020/04/17 | 0.0293 | 0.003 | 2,3 |
| 49 | Gaotai Lake | 35.41 | 90.96 | 10.59 | 2006/03/24-2021/04/15 | 0.0066 | 0.335 | 2,3 |
| 50 | Gasi Kule Lake | 38.12 | 90.79 | 115.81 | 2003/11/10-2021/06/25 | -0.0412 | < 0.001 | 1,2,3,7 |
| 51 | Gemang Co | 31.58 | 87.28 | 62.28 | 2009/10/01-2021/07/27 | 0.1551 | < 0.001 | 2,3 |
| 52 | Gemu Caka | 33.67 | 85.81 | 70.52 | 2003/10/16-2020/08/22 | -0.0314 | < 0.001 | 1,3,7 |
| 53 | Gopug Co | 31.86 | 83.18 | 61.63 | 2003/07/25-2020/06/09 | 0.0957 | < 0.001 | 1,2,3,7 |

| 54 | Goren Co | 31.12 | 88.35 | 478.16 | 2003/06/27-2021/07/25 | 0.1081 | < 0.001 | 1,2,3,5,7,8 |
|----|----------|-------|-------|--------|------------------------|--------|---------|-------------|
| 55 | Gouren Lake | 34.6 | 92.45 | 31.3 | 2005/06/02-2021/05/31 | 0.2184 | < 0.001 | 2,3,7 |
| 56 | Gozha Co | 35.02 | 81.07 | 245.34 | 2003/11/13-2021/07/15 | -0.0027 | < 0.001 | 1,3,7,8 |
| 57 | Guboke Co | 33.08 | 82.03 | 11.98 | 2004/01/18-2021/03/14 | -0.0175 | 0.007 | 1,3,7 |
| 58 | Guojialun Lake | 31.99 | 88.69 | 88.19 | 2010/12/23-2021/07/25 | 0.2775 | < 0.001 | 1,3,8 |
| 59 | Gyarab Punco | 32.2 | 87.78 | 51.9 | 2006/11/17-2021/03/28 | 0.0064 | < 0.001 | 2,3,5,6 |
| 60 | Gyaring Lake | 34.93 | 97.26 | 526 | 2007/10/04-2021/07/10 | 0.0276 | 0.055 | 1,2,3,7,8 |
| 61 | Gyesar Co | 30.21 | 84.8 | 142.1 | 2007/06/07-2017/05/06 | 0.1694 | 0.003 | 1,4,5 |
| 62 | Haidingnuo'er | 35.57 | 93.17 | 67.59 | 2010/11/03-2021/07/17 | -0.0977 | < 0.001 | 3,5 |
| 63 | Har Lake | 38.29 | 97.59 | 609.04 | 2003/09/18-2021/07/10 | 0.1894 | < 0.001 | 1,2,3,7,8 |
| 64 | Heishi North Lake | 35.56 | 82.74 | 112.4 | 2003/03/26-2021/04/28 | 0.3899 | < 0.001 | 1,2,3,5,7,8 |
| 65 | Hoh Xil Lake | 35.59 | 91.14 | 350.38 | 2005/06/20-2021/07/22 | 0.4792 | < 0.001 | 2,3,4,5,8 |
| 66 | Hot Spring Lake | 34.43 | 83.56 | 11.65 | 2008/03/08-2021/05/21 | 0.0033 | < 0.001 | 2,3 |
| 67 | Hulu Lake | 34.42 | 91.03 | 36.91 | 2011/11/08-2021/07/22 | 0.1848 | < 0.001 | 3,7 |
| 68 | Jiamucheng Co | 33.74 | 90.64 | 34.27 | 2007/03/11-2020/09/07 | 0.1651 | < 0.001 | 2,3,8 |
| 69 | Jiang Co | 31.55 | 90.82 | 40.48 | 2007/10/24-2021/01/30 | 0.1349 | 0.063 | 2,3 |
| 70 | Jiangchai Co | 32.16 | 90.46 | 28.64 | 2003/07/10-2021/06/27 | -0.0137 | 0.050 | 1,3,7 |
| 71 | Jidaocuo Lake | 32.52 | 83.22 | 12.76 | 2006/11/02-2020/12/02 | -0.0742 | 0.033 | 2,3 |
| 72 | Jieyue Lake | 35.07 | 90.27 | 17.76 | 2008/12/06-2020/06/21 | -0.0071 | 0.861 | 2,3 |
| 73 | Jieze Caka | 33.95 | 80.9 | 114.33 | 2003/12/18-2020/02/19 | 0.0725 | < 0.001 | 1,2,3,7,8 |
| 74 | Jingyu Lake | 36.33 | 89.44 | 339.57 | 2003/08/01-2021/07/25 | 0.4282 | < 0.001 | 1,3,7 |
| 75 | Jiuru Co | 31.01 | 89.92 | 39.95 | 2007/05/18-2020/12/03 | 0.0114 | 0.392 | 3,4,5,6 |
| 76 | Katiao Co | 33.96 | 82.97 | 61.09 | 2007/03/19-2021/04/07 | 0.7727 | < 0.001 | 2,3 |
| 77 | Kekao Lake | 35.7 | 91.36 | 74.39 | 2004/05/22-2021/06/27 | 0.4040 | < 0.001 | 1,2,3,4,5,7 |
| 78 | Kongmu Co | 29.01 | 90.45 | 36.94 | 2007/10/16-2021/04/15 | -0.0412 | < 0.001 | 2,3 |
| 79 | Kunggyu Co | 30.64 | 82.13 | 55.57 | 2004/05/18-2020/12/28 | 0.0600 | < 0.001 | 1,2,3,7,8 |
| 80 | Kunzhong Co | 33.1 | 80.39 | 13.77 | 2009/08/07-2021/01/27 | -0.0868 | 0.668 | 3,4,5 |
| 81 | Kusai Lake | 35.73 | 92.87 | 326.8 | 2002/09/29-2021/07/15 | 0.6215 | < 0.001 | 1,2,3,4,5,6,7,8 |

| 82 | Kushuihuan | 35.99 | 90.12 | 34.7 | 2004/04/17-2020/10/24 | -0.0234 | < 0.001 | 1,2,3,7 |
|---|---|---|---|---|---|---|---|---|
| 83 | Kyebxang Co | 32.45 | 89.98 | 187.11 | 2005/11/03-2021/06/30 | 0.2350 | < 0.001 | 2,3,8 |
| 84 | Lagkor Co | 32.03 | 84.13 | 95.62 | 2007/10/18-2021/04/26 | 0.1839 | < 0.001 | 2,3,8 |
| 85 | Langa Co | 30.69 | 81.23 | 256.24 | 2002/07/18-2020/11/23 | -0.1559 | < 0.001 | 1,3,4,5,6,7,8 |
| 86 | Langqiang Co | 28.72 | 85.88 | 24.03 | 2004/04/05-2021/03/06 | -0.0552 | < 0.001 | 1,3,7 |
| 87 | Laxiang Co | 33.98 | 86.04 | 25.46 | 2011/08/06-2021/07/30 | 0.2270 | < 0.001 | 1,3 |
| 88 | Laxiong Co | 34.34 | 85.23 | 66.92 | 2011/05/26-2021/07/07 | 0.3449 | < 0.001 | 1,3,7 |
| 89 | Lexiewudan Co | 35.75 | 90.2 | 273.3 | 2004/01/03-2021/07/23 | 0.5863 | < 0.001 | 1,2,3,4,5,7,8 |
| 90 | Lianhu Lake | 35.56 | 90.22 | 47.11 | 2007/04/03-2021/07/23 | 0.3035 | < 0.001 | 2,3,4,5 |
| 91 | Longmucuo Lake | 34.66 | 80.69 | 10.85 | 2004/08/03-2021/05/28 | 0.1170 | < 0.001 | 1,3,7 |
| 92 | Longre Co | 34.87 | 98.02 | 17.77 | 2003/09/03-2020/11/29 | 0.0227 | < 0.001 | 1,3,7 |
| 93 | Longwei Co | 33.87 | 88.31 | 57.85 | 2008/03/18-2021/06/30 | 0.2201 | < 0.001 | 2,3,7,8 |
| 94 | Lumajiangdong Co | 34.02 | 81.61 | 384.67 | 2003/07/12-2021/05/28 | 0.3865 | < 0.001 | 1,2,3,4,5,7,8 |
| 95 | Luotuo Lake | 34.44 | 81.94 | 68.22 | 2007/03/14-2021/04/08 | 0.2726 | < 0.001 | 1,2,3,4,5,8 |
| 96 | Maindung Co | 33.53 | 78.91 | 57.8 | 2006/02/26-2021/07/22 | -0.0370 | < 0.001 | 2,3,8 |
| 97 | Mang Co1 | 29.53 | 98.84 | 18.28 | 2004/01/05-2016/05/18 | 0.4612 | < 0.001 | 1,7 |
| 98 | Mapam Yumco | 30.68 | 81.47 | 412.69 | 2003/04/13-2021/07/15 | -0.0130 | 0.029 | 1,2,3,7 |
| 99 | Margai Caka | 35.12 | 86.75 | 158.05 | 2006/03/13-2021/04/20 | 0.6346 | < 0.001 | 2,3,8 |
| 100 | Margog Caka | 33.86 | 87.01 | 90.43 | 2009/01/22-2021/07/27 | 0.0328 | < 0.001 | 3,4,5,6 |
| 101 | Mazhangcuoqin | 34.34 | 91.59 | 67.93 | 2008/10/11-2021/01/05 | -0.0210 | 0.598 | 2,3 |
| 102 | Meiriqiecuomari | 33.64 | 89.72 | 97.18 | 2006/11/08-2021/05/10 | 0.2160 | < 0.001 | 2,3,7,8 |
| 103 | Memar Co | 34.22 | 82.31 | 166.67 | 2003/10/06-2021/07/12 | 0.4979 | < 0.001 | 1,2,3,7 |
| 104 | Mingjing Lake | 35.07 | 90.57 | 124.26 | 2003/09/01-2021/06/05 | 0.4452 | < 0.001 | 1,3,4,5,7,8 |
| 105 | Mudidalayu Co | 30.58 | 88.59 | 24.01 | 2004/09/04-2021/07/25 | 0.1051 | 0.001 | 1,3,7 |
| 106 | Mugqu Co | 31.06 | 89 | 78.04 | 2007/10/19-2021/06/07 | -0.0147 | 0.304 | 2,3,7 |
| 107 | Mushicuo Lake | 32.73 | 86.99 | 16.23 | 2004/04/05-2021/07/05 | 0.2426 | < 0.001 | 1,2,3,7,12 |
| 108 | Naka Co | 31.86 | 89.79 | 29.6 | 2007/10/28-2021/03/24 | 0.0576 | 0.087 | 2,3 |
| 109 | Nam Co | 30.74 | 90.6 | 2024.21 | 2003/03/08-2021/07/22 | 0.0305 | < 0.001 | 1,2,3,7,8 |

| 110 | Nariyong Co | 28.3 | 91.95 | 23.18 | 2013/09/26-2016/04/27 | 0.4404 | 0.115 | 7 |
| 111 | Nawu Lake | 32.93 | 82.08 | 20.06 | 2003/03/09-2020/04/15 | -0.0448 | < 0.001 | 1,3,7 |
| 112 | Ngangla Ringco | 31.54 | 83.08 | 492.8 | 2003/03/06-2021/06/19 | 0.0452 | < 0.001 | 1,2,3,4,5,7,8 |
| 113 | Ngangze Co | 31.02 | 87.13 | 471.6 | 2002/07/30-2021/06/10 | 0.2209 | < 0.001 | 1,2,3,4,5,6,7 |
| 114 | Ngoring Lake | 34.9 | 97.7 | 610 | 2002/06/10-2021/05/21 | 0.1363 | < 0.001 | 3,4,5,6,7,8 |
| 115 | Norma Co | 32.38 | 88.04 | 90.05 | 2007/11/13-2021/07/03 | 0.2758 | < 0.001 | 1,3,4,5,6,7 |
| 116 | Orba Co | 34.53 | 81.04 | 92.36 | 2003/04/17-2020/11/27 | 0.0093 | 0.275 | 1,3,4,5,6,8 |
| 117 | Paiku Co | 28.89 | 85.59 | 272.95 | 2005/11/08-2021/07/07 | -0.0967 | < 0.001 | 2,3,4,5 |
| 118 | Palung Co | 30.89 | 83.58 | 144.65 | 2004/05/15-2021/07/10 | 0.0676 | < 0.001 | 1,3 |
| 119 | Pipa Lake | 34.2 | 87.8 | 16.86 | 2012/05/06-2021/04/21 | 0.1973 | < 0.001 | 3 |
| 120 | Pongyin Co | 32.9 | 88.2 | 75.59 | 2010/03/21-2021/07/25 | 0.0934 | < 0.001 | 1,3,5,6 |
| 121 | Puma Yumco | 28.57 | 90.4 | 290.43 | 2006/03/08-2021/05/10 | -0.0568 | < 0.001 | 1,2,3,7 |
| 122 | Qiagang Co | 33.23 | 88.39 | 47.54 | 2005/10/31-2021/06/30 | 0.1137 | < 0.001 | 2,3,5,6 |
| 123 | Qiagui Co | 31.82 | 88.25 | 88.97 | 2004/01/22-2021/05/15 | -0.0138 | 0.023 | 1,2,3,6,7 |
| 124 | Qingche Lake | 34.48 | 81.79 | 71.51 | 2004/01/03-2021/05/03 | 0.2690 | < 0.001 | 1,3,7 |
| 125 | Qinghai Lake | 36.89 | 100.2 | 4348.25 | 2002/11/23-2021/07/02 | 0.1896 | < 0.001 | 1,2,3,4,5,7,8 |
| 126 | Qiongjiang Lake | 36.02 | 88.52 | 37.06 | 2007/03/18-2020/12/06 | 0.5334 | < 0.001 | 2,3,5,6 |
| 127 | Qoiden Co | 34.37 | 87.49 | 27.52 | 2003/10/13-2020/10/08 | -0.0308 | 0.025 | 1,3,7 |
| 128 | Quemo Co | 33.89 | 91.19 | 98.48 | 2008/03/17-2021/06/27 | 0.1993 | < 0.001 | 2,3,4,5,7 |
| 129 | Rebang Co | 33.03 | 80.58 | 46.22 | 2003/05/21-2021/05/05 | 0.0351 | < 0.001 | 1,3,7,8 |
| 130 | Rigain Punco | 32.58 | 86.24 | 42.79 | 2003/08/22-2021/03/29 | 0.0996 | < 0.001 | 1,3,7,8 |
| 131 | Rijiu Co | 34.2 | 91.7 | 13.26 | 2007/10/06-2021/05/05 | 0.0173 | 0.103 | 2,3 |
| 132 | Ringco Kongma | 30.93 | 89.67 | 138.48 | 2008/12/09-2021/07/23 | -0.0072 | < 0.001 | 2,3 |
| 133 | Rinqin Xubco | 31.28 | 83.45 | 186.55 | 2007/10/09-2021/04/30 | 0.1924 | < 0.001 | 2,3 |
| 134 | Rola Co | 35.44 | 88.41 | 169.9 | 2003/05/06-2021/07/03 | 0.2018 | < 0.001 | 1,3,7 |
| 135 | Salt Water Lake | 35.28 | 83.07 | 211.98 | 2008/02/28-2021/07/10 | 0.3751 | < 0.001 | 1,2,3,8 |
| 136 | Selin Co | 31.81 | 88.99 | 2300.37 | 2003/03/23-2021/07/25 | 0.3045 | < 0.001 | 1,2,3,4,5,7,8 |
| 137 | Serbug Co | 32 | 88.22 | 92.9 | 2003/08/01-2021/07/25 | 0.3072 | < 0.001 | 1,2,3,7 |

| 138 | Shibu Co | 31.39 | 88.73 | 14.1 | 2008/12/15-2020/11/23 | -0.0802 | < 0.001 | 2,3 |
|-----|----------|-------|-------|------|----------------------|---------|---------|-----|
| 139 | Shuanghu | 34.47 | 83.16 | 14.47 | 2013/03/24-2021/07/13 | 0.1593 | < 0.001 | 3 |
| 140 | Shuanglian Lake | 35.5 | 88.31 | 48.58 | 2011/04/05-2021/07/03 | 0.4009 | < 0.001 | 3,4,5 |
| 141 | Sugan Lake | 38.87 | 93.88 | 107.54 | 2003/02/16-2021/06/24 | 0.1075 | < 0.001 | 1,2,3,5,7 |
| 142 | Suona Lake | 33.92 | 86.69 | 27.03 | 2007/10/16-2021/05/15 | 0.0087 | < 0.001 | 2,3 |
| 143 | Tangra Yumco | 31.07 | 86.61 | 848.96 | 2003/09/08-2021/07/30 | 0.2117 | < 0.001 | 1,2,3,7,8 |
| 144 | Taro Co | 31.14 | 84.12 | 484.65 | 2007/10/18-2021/06/17 | 0.0439 | 0.048 | 2,3,4,5,7 |
| 145 | terang Punco | 33.06 | 89.07 | 32.52 | 2011/09/18-2021/07/02 | 0.0957 | < 0.001 | 1,3 |
| 146 | Tomgo Co | 31.72 | 86.98 | 24.08 | 2011/07/28-2020/09/13 | 0.0008 | 0.653 | 1,3 |
| 147 | Tso moriri | 32.9 | 78.32 | 142.54 | 2007/04/10-2021/06/04 | -0.1024 | < 0.001 | 2,3,4,5 |
| 148 | Tu Co | 33.4 | 89.86 | 448.23 | 2006/11/08-2021/07/23 | 0.4267 | < 0.001 | 2,3,4,5,7,8 |
| 149 | Tuoheping Co | 34.18 | 83.15 | 56.53 | 2003/04/30-2021/07/13 | -0.0565 | < 0.001 | 1,3,7,37 |
| 150 | Urru Co | 31.72 | 88 | 356.57 | 2003/05/23-2021/07/03 | 0.0314 | < 0.001 | 1,2,3,7 |
| 151 | Wanquan Lake | 34.24 | 83.81 | 67.42 | 2003/07/25-2021/04/30 | -0.1393 | < 0.001 | 1,3,7,8 |
| 152 | Weishan Lake | 35.96 | 89.24 | 46.83 | 2007/03/26-2021/07/25 | 0.2201 | < 0.001 | 2,3 |
| 153 | Wulanwula Lake | 34.8 | 90.48 | 651 | 2018/03/29-2021/07/27 | 0.2787 | 0.019 | 3 |
| 154 | Xiaga Co | 32.31 | 83.81 | 22.15 | 2008/02/28-2021/05/01 | 0.1346 | < 0.001 | 2,3,8 |
| 155 | Xiajian Lake | 34.16 | 82.77 | 13.92 | 2018/09/21-2021/04/03 | 0.0784 | 0.168 | 3 |
| 156 | Xiangyang Lake | 35.8 | 89.42 | 121.01 | 2007/10/03-2021/06/30 | 0.4468 | < 0.001 | 2,3,8 |
| 157 | Xianhe Lake | 36 | 88.07 | 50.71 | 2014/02/19-2021/05/13 | 0.4875 | < 0.001 | 3,7 |
| 158 | Xiaokusai Lake | 36.09 | 92.79 | 20.05 | 2013/07/12-2020/09/19 | -0.0069 | 0.002 | 3,8 |
| 159 | Xiasa'er Co | 31.58 | 80.99 | 13.83 | 2003/06/24-2021/07/23 | 0.0100 | < 0.001 | 1,2,3,4,5,7,8 |
| 160 | Xijir Ulan Lake | 35.21 | 90.34 | 462.69 | 2014/05/13-2021/07/07 | 0.3289 | 0.003 | 3 |
| 161 | Xuelian Lake | 34.09 | 90.26 | 54.06 | 2013/02/09-2021/07/27 | 0.1273 | < 0.001 | 3,7 |
| 162 | Xuguo Co | 31.95 | 90.34 | 35.07 | 2013/05/10-2021/07/30 | 0.0169 | < 0.001 | 3,7,8 |
| 163 | Yaggain Co | 31.56 | 89.01 | 112.39 | 2006/11/08-2021/07/23 | 0.9582 | < 0.001 | 2,3 |
| 164 | Yamzho Yumco | 28.96 | 90.71 | 548.29 | 2013/05/21-2021/06/24 | -0.2006 | < 0.001 | 3,8 |
| 165 | Yanghong Lake | 35.25 | 89.96 | 88.38 | 2007/04/08-2021/06/17 | 0.2879 | < 0.001 | 2,3,5,8 |

| 166 | Yanghu Lake | 35.41 | 84.59 | 163.09 | 2003/10/06-2021/07/23 | 0.6811 | 0.053 | 1,3,7,8 |
|-----|-------------|-------|-------|--------|------------------------|--------|-------|---------|
| 167 | Yangnapeng Co | 32.33 | 89.77 | 17.41 | 2003/11/13-2021/07/25 | 0.0410 | < 0.001 | 1,3,7 |
| 168 | Yanjian Lake | 34.77 | 89.03 | 18.19 | 2015/12/20-2020/01/02 | 0.5182 | 0.328 | 3 |
| 169 | Yinbo Lake | 36.19 | 88.14 | 50.01 | 2008/02/18-2021/06/05 | 0.4428 | < 0.001 | 2,3,8 |
| 170 | Yinlong Co | 33.91 | 88.04 | 17.5 | 2007/03/06-2021/07/25 | 0.2017 | < 0.001 | 1,3 |
| 171 | Yinma Lake | 35.6 | 90.63 | 105.23 | 2003/05/20-2021/06/05 | -0.1593 | < 0.001 | 1,3,7,8 |
| 172 | Yishan Lake | 35.24 | 90.91 | 27.61 | 2009/10/01-2021/07/20 | 0.2571 | < 0.001 | 2,3 |
| 173 | Yongbo Lake1 | 35.74 | 86.69 | 79.71 | 2004/03/07-2021/06/12 | 0.6614 | < 0.001 | 2,3 |
| 174 | Youyi Lake | 34.46 | 88.74 | 10.64 | 2007/03/18-2020/05/27 | 0.0085 | 0.105 | 2,3 |
| 175 | Yuan Lake1 | 34.81 | 89.29 | 17.22 | 2004/05/09-2021/07/02 | 0.1526 | < 0.001 | 3,4,5,6 |
| 176 | Yueliang Lake1 | 35.61 | 90.36 | 32.51 | 2007/10/28-2021/06/30 | 0.2762 | < 0.001 | 2,3 |
| 177 | Yulin Lake | 35.97 | 88.47 | 12.82 | 2008/10/10-2020/10/30 | 0.4261 | < 0.001 | 2,3 |
| 178 | Yuye Lake | 36.01 | 88.78 | 146.91 | 2003/10/29-2021/07/23 | 0.2125 | < 0.001 | 1,2,3,4,5 |
| 179 | Zhaliwa Co | 34.42 | 92.45 | 7.09 | 2013/01/05-2021/05/28 | 0.1126 | 0.058 | 3,7 |
| 180 | Zhamucuomaqiong | 33.15 | 89.7 | 30.7 | 2010/07/15-2021/04/18 | 0.0523 | 0.054 | 3,4,5 |
| 181 | Zhaoyang Lake | 35.3 | 87.26 | 92.28 | 2007/10/24-2021/07/05 | 0.0408 | < 0.001 | 1,2,3 |
| 182 | Zhari Namco | 30.93 | 85.61 | 1000.57 | 2002/08/02-2021/07/07 | 0.1671 | < 0.001 | 1,2,3,4,5,6,7,8 |
| 183 | Zhegucuo | 28.68 | 91.68 | 55.8 | 2003/10/01-2019/11/03 | 0.2692 | < 0.001 | 1,3,7 |
| 184 | Zhenquan Lake | 35.93 | 86.89 | 128.23 | 2004/06/06-2019/08/01 | 0.2616 | < 0.001 | 2,3,7,8 |
| 185 | Zige Tangco | 32.08 | 90.86 | 238.31 | 2002/08/01-2021/07/22 | 0.2200 | < 0.001 | 2,3,4,5,6,7 |
| 186 | Zigu Co | 31.37 | 87.9 | 76.17 | 2007/04/03-2021/07/03 | 0.0164 | 0.945 | 2,3 |
| 187 | Qagong Co | 34.44 | 82.33 | 30.73 | 2012/02/21-2021/06/20 | 0.3455 | < 0.001 | 3 |
| 188 | S63005 | 35.95 | 90.83 | 10.99 | 2013/12/05-2019/09/13 | -0.0871 | < 0.001 | 7,8 |
| 189 | Shen Co | 31.01 | 90.49 | 51.86 | 2014/06/17-2021/04/15 | -0.1095 | < 0.001 | 3,7,8 |
| 190 | Yaggain Co1 | 33.01 | 89.8 | 158.75 | 2013/09/16-2020/11/23 | 0.1842 | < 0.001 | 3,7,8 |
| 191 | Zhangnai Co | 31.54 | 87.4 | 43.98 | 2003/03/29-2021/03/26 | 0.1552 | < 0.001 | 1,2,3,5,6,7 |
| 192 | Zhaxi Co | 32.2 | 85.12 | 49.56 | 2010/03/11-2021/06/15 | 0.0760 | < 0.001 | 3,5,6,7 |
| 193 | Aiyong Co | 33.36 | 80.56 | 21.56 | 2015/02/19-2019/12/28 | 0.0862 | < 0.001 | 3,7 |

| 194 | Alake Lake | 35.57 | 97.12 | 34.7 | 2014/04/08-2021/06/12 | -0.0268 | 0.026 | 3,7 |
|---|---|---|---|---|---|---|---|---|
| 195 | Amjog Co | 29.63 | 86.25 | 22.01 | 2010/11/18-2021/04/23 | 0.0266 | < 0.001 | 3 |
| 196 | Angdar Co | 32.71 | 89.58 | 66.05 | 2013/10/21-2021/06/04 | 0.1110 | < 0.001 | 3 |
| 197 | Ayonggama Co | 34.78 | 98.29 | 14.38 | 2011/11/08-2021/06/10 | 0.0074 | 0.767 | 1,3 |
| 198 | Ayongwu'erma Co | 34.79 | 98.2 | 37.6 | 2016/07/16-2021/03/08 | 0.1736 | 0.004 | 3 |
| 199 | Baibing Lake | 35.9 | 86.42 | 21.87 | 2014/11/03-2021/05/15 | 0.6140 | < 0.001 | 3 |
| 200 | Baidoi Co | 32.79 | 87.83 | 79.17 | 2013/10/26-2021/07/27 | 0.2175 | < 0.001 | 3,7 |
| 201 | Bairab Co | 35.03 | 83.13 | 135.22 | 2012/03/21-2021/07/13 | -0.0165 | 0.143 | 3,8 |
| 202 | Baitan Lake | 34.56 | 88.58 | 20.13 | 2016/08/07-2021/05/11 | 0.0305 | 0.005 | 3 |
| 203 | Baitutang Lake | 34.65 | 87.61 | 10.4 | 2017/06/20-2020/06/27 | 0.0890 | < 0.001 | 3 |
| 204 | Bajiu Co | 28.79 | 90.85 | 30.2 | 2011/10/13-2018/11/30 | 0.2304 | < 0.001 | 3 |
| 205 | Bamco | 31.27 | 90.58 | 255.29 | 2012/11/13-2021/07/22 | -0.1468 | < 0.001 | 3 |
| 206 | Bandao Lake | 34.17 | 88.44 | 48.78 | 2014/11/23-2021/06/09 | 0.3673 | < 0.001 | 3,8 |
| 207 | Bei Hulsan Lake | 36.88 | 95.91 | 130.5 | 2012/08/06-2021/06/15 | 0.0026 | 0.161 | 3,7 |
| 208 | Beilei Co | 32.9 | 88.44 | 29.13 | 2018/06/17-2021/05/15 | 0.1563 | < 0.001 | 3 |
| 209 | Beiyu Lake | 33.03 | 86.18 | 15.1 | 2016/12/28-2019/08/27 | 1.0599 | < 0.001 | 8 |
| 210 | Bengze Co | 32.08 | 88.67 | 16.46 | 2010/11/10-2021/06/05 | 0.1052 | < 0.001 | 3 |
| 211 | Bero Zeco | 32.43 | 82.93 | 35.99 | 2013/06/17-2021/07/13 | 0.2038 | < 0.001 | 3,7 |
| 212 | Biluo Co | 32.9 | 88.84 | 35.12 | 2015/07/08-2021/07/25 | -0.0168 | 0.600 | 3 |
| 213 | Botao Lake | 34.01 | 89.96 | 71.36 | 2013/09/23-2021/07/23 | -0.0247 | 0.755 | 3 |
| 214 | Caiji Co | 31.21 | 85.44 | 33.05 | 2013/06/25-2021/04/25 | 0.1456 | < 0.001 | 3,7 |
| 215 | Caka Salt Lake | 36.7 | 99.11 | 115.77 | 2012/08/26-2021/06/09 | -0.0047 | 0.216 | 3,8 |
| 216 | Chabyer Co | 31.38 | 84.04 | 258.5 | 2012/08/07-2021/06/17 | 0.0256 | 0.021 | 3,7 |
| 217 | Chacang Co | 30.23 | 88.58 | 19.17 | 2014/07/08-2021/03/26 | 0.0254 | 0.005 | 3,7 |
| 218 | Chamu Co | 33.26 | 83.01 | 12.06 | 2016/04/02-2021/07/13 | 0.1501 | < 0.001 | 3 |
| 219 | Chanacuo Lake | 33.28 | 84.02 | 10.98 | 2016/09/16-2021/07/12 | 0.1064 | 0.534 | 3 |
| 220 | Chen Co | 28.95 | 90.52 | 39.4 | 2018/09/08-2021/04/15 | -0.3488 | < 0.001 | 3 |
| 221 | Co Ngoin2 | 31.47 | 91.5 | 84.86 | 2014/05/01-2021/04/13 | -0.0841 | 0.005 | 3 |

| 222 | Como Chamling | 28.4 | 88.22 | 38.57 | 2014/10/28-2020/02/05 | -0.1518 | 0.405 | 3 |
|-----|---------------|------|-------|-------|-----------------------|---------|-------|---|
| 223 | Cuoga Lake | 33.1 | 80.29 | 10.06 | 2019/10/31-2021/07/17 | 0.0385 | 0.296 | 3 |
| 224 | Cuojia Lake | 31.99 | 91.37 | 20.79 | 2015/02/09-2021/05/08 | -0.0052 | 0.346 | 3 |
| 225 | Cuojiangqin | 33.99 | 92.83 | 15.54 | 2012/11/07-2021/04/12 | -0.0027 | 0.117 | 3 |
| 226 | Cuolaba'e'eadong | 35.43 | 95.42 | 13.88 | 2019/05/08-2021/06/20 | 0.0219 | 0.398 | 3 |
| 227 | Dachaidan Lake | 37.84 | 95.25 | 33.14 | 2017/02/09-2021/04/28 | -0.0363 | < 0.001 | 3 |
| 228 | Dazadizha Co | 32.87 | 87.12 | 19.87 | 2013/08/12-2020/07/25 | 0.2458 | < 0.001 | 3,7,8 |
| 229 | Derucuo Lake | 32.69 | 88.88 | 10.61 | 2016/07/10-2021/03/26 | -0.0456 | 0.005 | 3 |
| 230 | Dingjiamang Co | 29.65 | 85.74 | 10.01 | 2012/01/18-2021/07/07 | 0.0099 | < 0.001 | 3 |
| 231 | Dongmo Co | 32.3 | 86.57 | 12.34 | 2013/10/04-2021/04/23 | -0.0245 | 0.035 | 3,7 |
| 232 | Dongyue Lake | 34.38 | 89.21 | 29.37 | 2014/08/27-2021/07/02 | 0.3215 | < 0.001 | 3 |
| 233 | Duolangcuoguo Lake | 32.23 | 85.86 | 11.15 | 2013/06/09-2021/05/18 | 0.1093 | < 0.001 | 3,7 |
| 234 | Duoma Co | 32.96 | 84.46 | 14.84 | 2015/11/07-2020/09/19 | 0.1965 | < 0.001 | 3 |
| 235 | East taijiner Lake | 37.49 | 93.92 | 101.8 | 2012/12/06-2021/06/19 | 0.0893 | 0.001 | 3 |
| 236 | Ezong Co | 32.86 | 89.47 | 14.75 | 2016/08/07-2021/05/13 | 0.1058 | 0.007 | 3 |
| 237 | Fenxing Lake | 34.39 | 88.42 | 12.41 | 2016/11/27-2021/06/09 | 0.1610 | 0.101 | 3 |
| 238 | Gahai1 | 37.13 | 97.55 | 34.87 | 2016/05/21-2020/06/06 | 0.0820 | 0.003 | 3 |
| 239 | Galala Co | 34.49 | 97.73 | 22.43 | 2013/11/26-2021/04/26 | -0.0056 | 0.040 | 3 |
| 240 | Gangma Co | 33.83 | 84.34 | 14.31 | 2016/08/19-2021/06/17 | 0.2253 | < 0.001 | 3 |
| 241 | Ganongcuo Lake | 31.91 | 91.53 | 17.8 | 2015/12/20-2021/06/25 | 0.0029 | 0.022 | 3,8 |
| 242 | Garen Co | 30.77 | 84.95 | 65.48 | 2014/10/06-2021/05/20 | 0.0303 | 0.084 | 3,8 |
| 243 | Garkung Caka | 33.97 | 86.49 | 70 | 2013/10/01-2021/06/10 | 0.3461 | < 0.001 | 3,8 |
| 244 | Gomang Co | 31.22 | 89.2 | 115.73 | 2020/12/15-2020/12/15 | -0.1396 | < 0.001 | 3 |
| 245 | Guogen Co | 32.4 | 89.19 | 57.9 | 2014/11/23-2021/07/23 | 0.1168 | < 0.001 | 3,8 |
| 246 | Haobo Lake | 34.4 | 88 | 18.89 | 2017/11/07-2021/07/27 | 0.1476 | 0.015 | 3 |
| 247 | Hehua Lake | 36.14 | 88.99 | 29.49 | 2014/05/10-2021/06/29 | 0.8498 | < 0.001 | 3,8 |
| 248 | Heihai | 35.99 | 93.26 | 38.16 | 2011/08/04-2021/05/26 | 0.0706 | 0.405 | 1,3,13 |
| 249 | Hengliang Lake | 34.88 | 89.05 | 23.66 | 2013/09/28-2021/07/25 | 0.2989 | < 0.001 | 3,7 |

| 250 | Huangshui Lake | 34.33 | 87.7 | 31.29 | 2014/05/10-2019/12/13 | 0.2205 | < 0.001 | 3,8 |
|-----|----------------|-------|------|-------|------------------------|--------|---------|-----|
| 251 | Huolunuo'er | 35.56 | 91.93 | 160.15 | 2013/09/18-2021/07/17 | -0.1708 | < 0.001 | 3,8 |
| 252 | Jiaomu Caka | 33.27 | 87.22 | 25.52 | 2017/01/02-2019/08/04 | 0.2970 | < 0.001 | 8 |
| 253 | Jiaruo Co | 32.19 | 86.6 | 13.15 | 2014/05/03-2016/05/06 | 0.0651 | 0.850 | 7 |
| 254 | Kaba Niu'erduo | 35.42 | 95.11 | 29.08 | 2013/07/17-2021/06/17 | 0.0034 | 0.831 | 3 |
| 255 | Kahu Co | 33.39 | 82.97 | 30.56 | 2016/01/06-2020/01/21 | 0.1124 | 0.002 | 3 |
| 256 | Kanbakadong Co | 35.21 | 95.13 | 20.6 | 2017/01/12-2020/03/14 | 0.0080 | 0.296 | 3 |
| 257 | Kangru Caka | 33.56 | 86.96 | 15.49 | 2016/05/03-2020/08/19 | 0.0902 | < 0.001 | 3,8 |
| 258 | Keluke Lake | 37.28 | 96.89 | 54.55 | 2014/01/27-2021/05/23 | -0.0011 | 0.277 | 3 |
| 259 | Kong Co | 30.82 | 88.35 | 13.8 | 2019/05/26-2019/05/26 | 0.1380 | < 0.001 | 3 |
| 260 | Koucha | 34.01 | 97.23 | 17.5 | 2016/11/08-2019/11/18 | -0.0120 | 0.028 | 3 |
| 261 | Kuhai | 35.3 | 99.18 | 47.32 | 2014/12/25-2021/02/06 | 0.1000 | < 0.001 | 3 |
| 262 | Labu Co | 32.96 | 83.8 | 15.36 | 2017/07/23-2021/04/05 | 0.1938 | 0.905 | 3 |
| 263 | Lingguo Co | 33.85 | 88.6 | 125.8 | 2013/05/08-2021/05/15 | 0.6406 | < 0.001 | 3 |
| 264 | Ma'erxia Co | 30.97 | 87.47 | 102.07 | 2014/02/16-2021/04/21 | 0.1358 | < 0.001 | 3,8 |
| 265 | Mang Co2 | 34.49 | 80.44 | 12.92 | 2016/09/22-2019/10/01 | 0.1572 | 0.002 | 3 |
| 266 | meijuhu | 36.02 | 88.41 | 17.48 | 2013/07/28-2021/07/03 | 0.7254 | < 0.001 | 3,7,37 |
| 267 | Merqung Co | 31.02 | 84.58 | 60.27 | 2017/05/26-2021/07/08 | 0.1298 | < 0.001 | 3 |
| 268 | Monco Bunnyi | 30.64 | 86.26 | 150.78 | 2014/01/02-2021/05/18 | 0.0834 | 0.007 | 3,7 |
| 269 | Naiqam Co | 32.32 | 88.69 | 45.99 | 2014/01/17-2021/02/03 | 0.0154 | 0.003 | 3,7 |
| 270 | Nanzha Co | 32.66 | 85.47 | 25.1 | 2013/01/19-2020/09/16 | 0.1950 | < 0.001 | 3 |
| 271 | Neri Punco | 31.3 | 91.47 | 92.61 | 2013/02/03-2021/04/13 | -0.1365 | < 0.001 | 3,5 |
| 272 | Ngoinyar Coqung | 32.99 | 88.7 | 96.58 | 2013/10/24-2021/05/11 | 0.1414 | < 0.001 | 3 |
| 273 | Ningri Co | 33.32 | 85.58 | 16.42 | 2013/11/28-2020/10/10 | -0.1068 | 0.321 | 3,7 |
| 274 | Niri Acuogai | 33.09 | 93.21 | 35.29 | 2011/11/13-2021/07/15 | 0.0546 | 0.003 | 1,3 |
| 275 | Niudu Lake | 33.65 | 88.58 | 10.23 | 2010/12/03-2020/11/20 | 0.0498 | 0.014 | 3,5,6 |
| 276 | Noname | 33.16 | 89.34 | | 2017/12/01-2021/06/29 | -0.0583 | 0.325 | 3 |
| 277 | Nyer Co | 32.28 | 82.22 | 22.13 | 2019/01/17-2021/04/05 | 0.0852 | 0.276 | 3 |

| 278 | Pa Co | 31.91 | 90.04 | 13.43 | 2017/11/30-2020/07/19 | -0.2014 | 0.002 | 3 |
|-----|-------|-------|-------|-------|----------------------|---------|-------|---|
| 279 | Pozi Co | 30.47 | 86.11 | 25.66 | 2016/06/18-2021/01/15 | 0.1312 | < 0.001 | 3 |
| 280 | Puga Co | 31.11 | 89.55 | 43.43 | 2014/08/27-2020/08/15 | -0.1463 | 0.022 | 3,8 |
| 281 | Pur Co | 34.88 | 81.96 | 40.64 | 2016/01/08-2020/04/14 | -0.0078 | 0.136 | 3 |
| 282 | Pusai'er Co | 32.34 | 89.46 | 33.89 | 2013/07/20-2021/06/29 | 0.0591 | 0.103 | 3,7 |
| 283 | Puxu Co | 31.91 | 87.21 | 16.58 | 2016/11/04-2020/04/04 | -0.0486 | 0.026 | 3 |
| 284 | Qieli Co | 31.68 | 90.97 | 12.92 | 2010/09/10-2021/05/08 | -0.1155 | < 0.001 | 3 |
| 285 | Qige Co | 31.2 | 85.53 | 20.29 | 2016/09/13-2021/07/07 | 0.0074 | 0.684 | 3 |
| 286 | Qingwa Lake | 34.71 | 86.4 | 25.22 | 2017/04/26-2020/10/08 | 0.0794 | 0.016 | 3 |
| 287 | Qiuruba Lake | 33.31 | 84.81 | 10.65 | 2018/04/02-2020/07/04 | 0.2911 | 0.169 | 3 |
| 288 | Quanshui Lake | 34.76 | 80.18 | 16.74 | 2016/03/05-2021/07/20 | 0.2077 | 0.002 | 3,8 |
| 289 | Rejue Caka | 33.69 | 86.85 | 33.12 | 2013/08/12-2021/04/20 | 0.1031 | < 0.001 | 3,7 |
| 290 | Rena Co | 32.73 | 84.26 | 20.7 | 2016/05/18-2020/10/14 | 0.1002 | < 0.001 | 3,8 |
| 291 | Rige Co | 34.33 | 98.75 | 16.45 | 2016/04/27-2021/04/25 | -0.0039 | < 0.001 | 3,8 |
| 292 | Riju Co | 33.8 | 90.36 | 26.12 | 2013/02/07-2020/10/24 | 0.0472 | < 0.001 | 3 |
| 293 | Ringco Ogma | 30.93 | 89.84 | 66.92 | 2013/06/02-2021/06/30 | -0.2125 | < 0.001 | 3,7 |
| 294 | S54001 | 36.19 | 89.16 | 13.21 | 2011/09/08-2020/08/16 | 0.2422 | < 0.001 | 1,3 |
| 295 | S63008 | 35.95 | 89.33 | | 2016/10/29-2019/11/07 | 0.3496 | < 0.001 | 3 |
| 296 | S63022 | 35.23 | 91.21 | 13.75 | 2014/08/27-2021/03/21 | -0.0015 | < 0.001 | 3 |
| 297 | Sandao Lake | 34.73 | 83.88 | 32.81 | 2015/06/20-2020/08/26 | 0.4259 | < 0.001 | 3,8 |
| 298 | Sekezhi Co | 32 | 82.05 | 19.15 | 2021/04/30-2021/04/30 | 0.2214 | < 0.001 | 3 |
| 299 | Sengli Co | 30.44 | 84.06 | 83.29 | 2016/04/17-2020/09/19 | -0.0087 | 0.497 | 3,8 |
| 300 | Shengli Lake | 35.29 | 86.27 | 36.78 | 2015/08/12-2021/07/07 | 0.7865 | 0.003 | 3 |
| 301 | Shuangju Lake | 34.94 | 87.3 | 10.82 | 2019/04/30-2021/03/04 | 0.1057 | 0.351 | 3 |
| 302 | Shuixiang Lake | 36.03 | 87.88 | 15.52 | 2013/08/31-2021/05/16 | 0.4641 | < 0.001 | 3 |
| 303 | Sijia Lake | 34.04 | 82.61 | 24.62 | 2018/11/20-2021/05/23 | 0.2260 | 0.027 | 3 |
| 304 | Songmuxi Co | 34.61 | 80.25 | 30.8 | 2015/05/05-2020/02/24 | 0.1364 | 0.024 | 3 |
| 305 | T54001 | 34.22 | 89.75 | 19.1 | 2018/07/12-2020/07/19 | 0.1441 | 0.014 | 3 |

| 306 | T54024 | 34.92 | 81.69 | 17.99 | 2017/12/20-2021/04/30 | 0.3288 | 0.044 | 3 |
|-----|--------|-------|-------|-------|-----------------------|--------|-------|---|
| 307 | Taiping Lake | 34.3 | 89.71 | 28.48 | 2018/07/12-2020/09/09 | 0.1672 | < 0.001 | 3 |
| 308 | Taiyang Lake | 35.93 | 90.63 | 101.44 | 2013/10/11-2021/04/13 | -0.0005 | 0.853 | 3,7 |
| 309 | Tao Lake | 36.17 | 89.33 | 32.49 | 2015/10/26-2020/11/23 | 0.2384 | < 0.001 | 3 |
| 310 | Taoxing Lake | 33.88 | 84.02 | 10.52 | 2016/06/24-2021/04/02 | 0.0428 | 0.002 | 3 |
| 311 | Tari Co | 31.52 | 85.68 | 40.11 | 2014/04/17-2021/06/12 | -0.0147 | 0.045 | 3,8 |
| 312 | Telashi Lake | 34.81 | 92.22 | 73.65 | 2010/06/07-2021/06/22 | 0.3263 | < 0.001 | 3,5,6,8 |
| 313 | Tungpu Co | 31.31 | 87.23 | 32.95 | 2021/01/14-2021/01/14 | 0.2309 | < 0.001 | 3 |
| 314 | Tuosu Lake | 37.14 | 96.94 | 150.65 | 2011/07/03-2021/06/12 | 0.7239 | < 0.001 | 1,3 |
| 315 | Tuzhong Lake | 34.53 | 84.7 | 32.28 | 2015/03/24-2020/09/19 | 0.3202 | < 0.001 | 3 |
| 316 | Wan'an Lake | 34.43 | 88.55 | 19.87 | 2013/08/29-2020/05/25 | 0.0894 | < 0.001 | 3,8 |
| 317 | Wandou Lake | 34.56 | 90.85 | 22.81 | 2013/09/21-2021/04/15 | 0.1611 | < 0.001 | 3,5 |
| 318 | Wuga Co | 32 | 86.65 | 11.56 | 2019/12/15-2021/04/23 | 0.0122 | 0.865 | 3 |
| 319 | Wujiongcuo Lake | 30.91 | 86.42 | 14.66 | 2003/09/01-2021/06/30 | 0.7219 | < 0.001 | 1,2,3,4,5,7,8 |
| 320 | Xiabie Co | 32.22 | 87.27 | 20.71 | 2011/08/14-2021/05/21 | 0.1412 | < 0.001 | 1,3 |
| 321 | Xiangtao Lake | 34.13 | 84.97 | 11.31 | 2004/02/27-2021/07/02 | 0.4246 | < 0.001 | 1,2,3,7 |
| 322 | Xiao Caka | 33.06 | 87.78 | 28.2 | 2013/06/14-2021/07/13 | 0.3508 | < 0.001 | 3 |
| 323 | Xiaosugan Lake | 39.07 | 94.21 | 11.87 | 2008/12/12-2019/08/31 | 0.0230 | 0.427 | 2,3,8 |
| 324 | Xiligou Lake | 36.84 | 98.46 | 43.31 | 2016/02/25-2021/07/02 | 0.0581 | < 0.001 | 3 |
| 325 | Xingbo Lake | 35.68 | 87.04 | 12.06 | 2013/10/31-2021/03/11 | 0.8170 | < 0.001 | 3 |
| 326 | Xinhu Lake | 34.39 | 84.25 | 61.04 | 2018/05/30-2020/11/02 | 0.3981 | 0.071 | 3 |
| 327 | Xinxin Lake | 34.83 | 98.11 | 26.28 | 2015/11/02-2020/01/06 | -0.0529 | < 0.001 | 3 |
| 328 | Xuejing Lake | 35.98 | 87.38 | 86.08 | 2003/10/06-2021/06/30 | 0.4701 | < 0.001 | 1,3,4,5,8 |
| 329 | Xuemei Lake | 36.29 | 88.27 | 56.26 | 2005/05/20-2021/07/22 | 0.6780 | 0.148 | 2,3 |
| 330 | Xuru Co | 30.29 | 86.42 | 210.03 | 2016/05/25-2021/06/19 | 0.1013 | 0.002 | 3 |
| 331 | Yadao Lake | 33.96 | 83.32 | 19.5 | 2006/06/10-2021/06/07 | 0.2246 | < 0.001 | 2,3,7 |
| 332 | Yaggain Co2 | 32.35 | 87.31 | 48.78 | 2018/01/02-2021/03/01 | 0.2445 | 0.006 | 3 |
| 333 | Yake Co | 34.7 | 87.19 | 20.41 | 2003/03/28-2021/07/22 | 1.5657 | < 0.001 | 1,2,3,4,5,7,8 |

| 334 | Yan Lake | 35.52 | 93.41 | 144.32 | 2003/11/30-2021/07/22 | 2.3845 | < 0.001 | 1,3,7 |
|-----|----------|-------|-------|--------|------------------------|--------|---------|-------|
| 335 | Yanzi Lake | 33.87 | 89.93 | 16.07 | 2016/06/02-2021/05/06 | 0.0199 | < 0.001 | 3 |
| 336 | Yaxi Co | 34.25 | 92.68 | 25.17 | 2017/10/13-2020/10/10 | 0.0239 | 0.053 | 3 |
| 337 | Woniu_Lake | 35.73 | 85.27 | 15.78 | 2013/10/04-2021/06/10 | 0.2039 | 0.108 | 3,5,7 |
| 338 | Yazi Lake | 35.07 | 87.07 | 44.24 | 2014/05/15-2021/07/30 | 0.5351 | < 0.001 | 3 |
| 339 | Yelusu Lake | 35.22 | 92.14 | 202.47 | 2011/11/08-2021/07/20 | 0.0342 | < 0.001 | 3,7,8 |
| 340 | Yibug Caka | 32.94 | 86.71 | 178.36 | 2013/11/23-2021/05/15 | 0.1059 | < 0.001 | 3,5 |
| 341 | Yingtian Lake | 34.43 | 88.06 | 16.93 | 2012/01/14-2021/03/28 | 0.1110 | < 0.001 | 3 |
| 342 | Yongbo Lake2 | 34.96 | 89.24 | 43.59 | 2010/04/25-2020/12/06 | 0.1208 | < 0.001 | 3,5,6 |
| 343 | Yoqag Co | 30.47 | 88.61 | 68.19 | 2016/07/10-2021/03/26 | 0.0016 | 0.635 | 3 |
| 344 | Youbu Co | 30.8 | 84.8 | 64.15 | 2013/09/21-2020/03/13 | 0.2190 | < 0.001 | 3,7 |
| 345 | Yuan Lake2 | 33.95 | 85.34 | 14.04 | 2018/04/28-2021/06/15 | 0.1111 | 0.384 | 3 |
| 346 | Yueliang Lake2 | 35.62 | 86.27 | 12.54 | 2013/05/13-2021/06/10 | -0.0433 | 0.567 | 3 |
| 347 | Yueya Lake | 34.92 | 82.22 | 14.21 | 2019/12/21-2021/07/15 | -0.0219 | < 0.001 | 3 |
| 348 | Yuhuan Lake | 34.8 | 83.92 | 17.28 | 2017/09/18-2021/04/02 | 0.3645 | 0.132 | 3,8 |
| 349 | Yupan Lake | 34.9 | 88.39 | 21.11 | 2015/12/28-2020/10/04 | 0.2744 | < 0.001 | 3 |
| 350 | Zainzong Co | 32.24 | 89.61 | 12.56 | 2016/02/16-2021/04/18 | -0.0586 | 0.068 | 3 |
| 351 | Zhangtoujiangmu Co | 35.33 | 95.61 | 17.84 | 2016/01/05-2021/05/26 | 0.0041 | 0.034 | 3 |
| 352 | Burog Co | 34.4 | 85.77 | 92.95 | 2016/03/07-2021/04/23 | 0.4081 | < 0.001 | 3,8 |
| 353 | Dongka Co | 31.78 | 90.4 | 72.5 | 2019/06/17-2021/07/22 | -0.2658 | 0.503 | 3 |
| 354 | Kongkong Caka | 33.16 | 88.11 | 49.52 | 2013/04/08-2021/07/25 | 0.0535 | < 0.001 | 3,8 |
| 355 | West taijiner Lake | 37.71 | 93.38 | 99 | 2014/02/01-2020/08/29 | -0.0112 | < 0.001 | 3 |
| 356 | Xiaochaidan Lake | 37.5 | 95.51 | 88.13 | 2009/06/24-2019/08/06 | 0.2674 | 0.001 | 3,4,5 |
| 357 | Laorie Co | 33.73 | 90.01 | 56.6 | 2007/04/21-2021/07/23 | -0.0284 | < 0.001 | 3,4,5,6 |
| 358 | Pung Co | 31.5 | 90.97 | 176.46 | 2003/07/25-2021/06/27 | 0.1797 | < 0.001 | 1,3,7 |
| 359 | Ciyijiare Lake | 32.61 | 87.21 | 10.05 | 2019/02/03-2021/07/27 | -0.1646 | 0.005 | 3,8 |
| 360 | Ma'an Lake | 35.23 | 89.51 | 18.55 | 2011/03/13-2021/06/07 | 0.0067 | 0.921 | 3,5,6 |
| 361 | Xuehuan Lake | 35.01 | 88.05 | 40.98 | 2012/02/09-2021/07/05 | 0.2427 | < 0.001 | 3 |

*altimeter type; 1 - Envisat, 2 – ICESat-1, 3 - CryoSat-2, 4 - Jason-1, 5 - Jason-2, 6 - Jason-3, 7 - SARAL, 8 - Sentinel-3A.