# Peer review of "A dataset for lake level changes in the Tibetan Plateau from 2002 to 2021 using multi-altimeter data"

_Earth System Science Data, 2022_

## Author Response (AR1)

**Responses to the comments on the Manuscript "High-resolution datasets for lake level changes in the Tibetan Plateau from 2002 to 2021 using multi-altimeter data"**

Dear editor,

The authors would like to express thanks to the anonymous reviewers for their voluntary work and the constructive comments to improve this manuscript. All of the comments are of great benefit to us. During the past few days, we did much work to revise the manuscript according to the reviewer's comments. All of the comments have been addressed. Our revisions are as follows.

**RC1**: 'Comment on essd-2022-313', Anonymous Referee #1, 17 Nov 2022

This study provides the time series of water level for lakes in the Qinghai-Tibetan Plateau between 2002 and 2021 using altimeter data from Envisat, ICESat-1, CryoSat-2, Jason-1, Jason-2, Jason-3, SARAL, and Sentinel-3A. The water level data in 2002-2021 provided by this study is not new, and have been reported by couple of previous studies. This study did not present well such as time series of lake level and not story focused such as the mention the discharge without close relation with study study. Moreover, the authors did not know the background information of lakes over the Tibetan Plateau well. I can not recommend the publication of this manuscript (dataset).

Reply: Although the water level data provided by this study is not new, we provide the largest number of monitored lakes among the available studies. In the revised version, we added the analysis of the variation of lake level in different basin, while the discharge mentioned in the manuscript is mainly to illustrate the application of lake level data, which refers to the fact that we can explore the regulation of the rivers associated with the lake using lake level data.

1) Although this study provides the time series of water level of lakes in the Qinghai-Tibetan Plateau from eight altimetry products, this dataset is not new compared with published studies, especially in a limited period (2002-2021). Hydroweb and other websites have provided open access lake level data since 1992, which has covered the altimetry data used in this study.

Reply: These eight altimetry products are already used in published studies, but the number of lakes monitored based on them is very limited, as can be seen in the following table.

| Reference | No. of Lakes | Period | Data Source | Dataset Public or not |
|---|---|---|---|---|

| | | | | |
|---|---|---|---|---|
| Jiang et al. (2017) | 70 | 2003-2015 | IceSat-1, Cryosat-2 | N |
| Zhang et al. (2017) | 68 | 1989-2015 | IceSat-1, Landsat | N |
| Li et al. (2017) | 167 | 2002-2012 | IceSat-1, Envisat | N |
| Hwang et al. (2019) | 59 | 2003-2016 | Jason-2/3, SARAL, IceSat-1, Cryosat-2 | N |
| Li et al. (2019) | 52 | 2000-2017 | Jason-1/2/3, Envisat, Cryosat-2, IceSat-1 | Y |
| Zhang et al. (2019) | 62 | 2003-2018 | IceSat-1/2 | Y |
| Hydroweb (Cretaux et al. 2011) | 36 | 1993-2022 | ERS-2, Envisat, T/P, IceSat-1, SARAL, Jason-1/2/3, Cryosat-2, Sentinel-3A | Y |
| DAHITI (Schwatke et al. 2015) | 62 | 2003-2022 | ERS-2, Envisat, SARAL, Sentinel-3A, Cryosat-2, IceSat-1, Jason-2/3, | Y |
| This Study | 361 | 2002-2021 | Envisat, SARAL, IceSat-1, Cryosat-2, Jason-1/2/3, Sentinel-3A | Y |

As we know, a large number of lakes exist on the Tibetan Plateau, and monitoring the water levels of these lakes is very important for understanding the water cycle on this plateau. Our study monitored the largest number of lakes with an area greater than 10 km$^2$ compared to previous studies. We believe that more monitored lakes will be useful to find more details driving mechanisms, and patterns of changes in the Tibetan Plateau. Therefore, this is also the objective of this manuscript.

2) The presentation of this study is not good such as Figures 3 and 4. Flowchart 1 and 5 should be combined together. For a scientific paper, the figures should be drawn by a scientific standard. Moreover, the offset among the different altimetry data was addressed? How?

Reply: Thank for your comments. We have combined the Figure 3 and 4, and drawn the figure again, please see the revised manuscript. Since Flowchart 1 and 5 represent different processes for altimetry data, Flowchart 1 is mainly the waveform retracking processing of altimetry data, while process 5 is mainly the fusion processing of water level data extracted from multi-source altimeters, which are different and not suitable for merging, so the two flowcharts are retained.

In this manuscript, the main merged method removes the offset between different altimetry data by subtracting the mean discrepancy obtained during the overlap period. Thus, we will pick the dynamic reference time series to make the mergerd time series

as long as possible for each time in case there are no overlap period.

On the other hand, it also exists some lakes with no overlap period when merging ICEsat-1 and Cryosat-2. In this case, we will consider using a combined linear-periodic-residual model (Liao et al., 2014) to simulate and forecast lake-level time series in the no-overlap period, and then make it possible to obtain the offset between ICEsat-1 and Cryosat-2. These details are presented in section 3.2.

3) What is the difference of boundaries between Qinghai-Tibetan Plateau and Tibetan Plateau? How the comparison of time series of altimetry data and in-situ? Why the streamflow and discharge data are used, but the analysis of water level and advantage of your study are not clear?

Reply: There is no difference of boundaries between Qinghai-Tibetan Plateau and Tibetan Plateau. In China, people used to call Qinghai-Tibetan Plateau (QTP), but internationally, it is used to call Tibetan Plateau (TP). To avoid misunderstanding, we have revised as the Tibetan Plateau in this manuscript.

Due to the unknown datum of in situ data, here we consider comparing the water level anomaly between in situ data and lake level in this study by removing the mean value over the validation period. In the revised vision, in order to show the validate results of lake level in this study, we added a figure comparing the in situ data with the lake level in this study.

The streamflow and discharge data are used to show that lake levels can be used to explore the regulation of the rivers associated with the lake, and this is just one case of the application of lake levels. In addition, we added the analysis of water level changes in lakes in different basin, so it further indicates that our study is much clear in reflecting the spatial and temporal variability of lakes on the Tibetan Plateau.

RC2: 'Comment on essd-2022-313', Anonymous Referee #2, 26 Dec 2022

- Line 13: Here is for global climate change. But the text cannot discuss the relation between lake level change and global climate change.

Reply: Thanks for the suggestions, but here we are not discussing the relation between lake level change and global climate change. Just an introduction for the Tibetan Plateau and describe the importance of lakes in Tibetan Plateau.

- Line 16: What about the detail altimetry satellite missions?

Reply: Thanks for the suggestions, the altimetry missions we used in this manuscript is added in the revised version, please see the revised manuscript.

- Line 19: Here the time spans 2002 to 2021. Are all lakes' levels in this time span?

  **Reply:** Thanks for the suggestions, not all the lakes staring from 2002, among all of this, 167 lakes water level staring from 2010. The details also added into revised version.

- Section 1: There are many literatures about the lake level changes in QTP, which should be further summarized and generalized.

  **Reply:** Thanks for the suggestions, we added related literatures in past 5 years and summarized in the table 1.

Table 1 Comparison of this study with previous studies

| Reference | No. of Lakes | Period | Data Source | Dataset Public or not |
|---|---|---|---|---|
| Jiang et al. (2017) | 70 | 2003-2015 | IceSat-1, Cryosat-2 | N |
| Zhang et al. (2017) | 68 | 1989-2015 | IceSat-1, Landsat | N |
| Li et al. (2017) | 167 | 2002-2012 | IceSat-1, Envisat | N |
| Hwang et al. (2019) | 59 | 2003-2016 | Jason-2/3, SARAL, IceSat-1, Cryosat-2 | N |
| Li et al. (2019) | 52 | 2000-2017 | Jason-1/2/3, Envisat, Cryosat-2, IceSat-1 | Y |
| Zhang et al. (2019) | 62 | 2003-2018 | IceSat-1/2 | Y |
| Hydroweb | 36 | 1993-2022 | ERS-2, Envisat, T/P, IceSat-1, SARAL, Jason-1/2/3, Cryosat-2, Sentinel-3A | Y |
| DAHITI | 62 | 2003-2022 | ERS-2, Envisat, SARAL, Sentinel-3A, Cryosat-2, IceSat-1, Jason-2/3, | Y |
| This Study | 361 | 2002-2021 | Envisat, SARAL, IceSat-1, Cryosat-2, Jason-1/2/3, Sentinel-3A | Y |

- Line 72: Here there are 364 lakes. But there are 262 lakes in line 19.

  **Reply:** Thanks for the suggestions, we have revised it, the correct number is 361. We also scrutiny the whole text, and revised this mistake.

- Table 1: One altimetry satellite can only pass some lakes. Some lakes can be covered by two or three altimetry satellite mission. How to process these conditions to precisely determine one level series for one lake? How to get the lake level series of all lakes from 2002 to 2021?

**Reply:** Thanks for the comments.

After constructing the time series for each altimetry over related lakes. We can follow the section 3.2 to fuse the multi-altimeter time series.

Specially, we first merged the two products with the longest period for the time series and chose the altimeter-derived water level with the longer time series as the baseline. Then systematic biases between another altimeter and the baseline will be removed by subtracting the mean discrepancy during the overlap period compared with the reference series (Lee et at., 2011; Kropáček et al., 2012) according to Eq. (4). Then, the same process was applied to the remaining products and the merged products connecting the three altimeters.

Additionally, some lakes cannot merge successfully (if just pass ICEsat-1 and Cryosat-2 before 2013), we will try to use a combined linear-periodic-residual model (Liao et al., 2014) to simulate and forecast the lake-level time series in the no-overlap period to merge the two altimeters with no overlap period.

We added some details to make it easier to follow in revised version.

Unfortunately, this manuscript only estimate lake level has valid observations from multi-altimetry, so not all the 361 lakes for time series from 2002 to 2021, only 194 lakes for the time series from 2002 to 2021, and 167 lakes for the time series from 2010 to 2021.

- Table 2: What unit is used for coordinate? 1985 should be the National Height Datum of China.

**Reply:** Thanks for the suggestions, the unit for coordinate is degree. We revised the table notes for coordinates and 1985. Please see the revised manuscript.

- (1): How to get these corrections? How to improve these corrections?

  **Reply:** Thanks for the comments. With the exception for retracking correction, all the corrections are included in the altimetry data product. We just need to improve the retracking corrections which is the most important to decide the accuracy of lake level. More detail can be seen in section 3.1.1.

- Section 3.1.1: How to verify the effectiveness and accuracy of the retracking method?

  **Reply:** Thanks for the suggestions. Actually, the retracking method we used here is the automatic multiscale-based peak detection retracker (AMPDR), having a special paper for it (Chen et al., 2021). We try to compare several lake level time series from altimetry with in-situ time series and existing product time series. The effectiveness and accuracy of this retracker has already discussed in detail in the paper above.

- Section 3.1.2: Here is noise footprints. How about the abnormal footprint?

  **Reply:** Thanks for the suggestions. The abnormal footprints are already detected in section 3.1.1, we call as unavailable off-nadir observation, which will be removed during retracking.

  The lake level from invalid off-nadir observation usually shows a large bias with the DistanceThres. So, if the difference is larger than the half of the range window, it may not provide the signals from water surface and this observation will be regarded as an invalid off-nadir point (using Sentinel-3 as an example, it is $1/2*128*0.4684$). (Chen et al., 2021)

- Section 3.1.3: How to process the gross errors in time series?

  **Reply:** Thanks for the suggestions. Here we used a state-space model (Nielsen et al., 2015), the gross errors can be obtained from their models. The obtained time series are not easily averaging, they considering continuous time steps and a hypothetical error model. Then according to the Laplace estimation, the mean value and error will be calculated from the model. More detail can be seen in Nielsen et al. (2015).

- Section 3.2: Coordinate system transfer and coordinate frame transfer all should be made.

  **Reply:** Thanks for the suggestions. Coordinate system transfer and coordinate frame transfer are all made in this manuscript. All the data will be transferred into WGS84/EGM 2008.

- (5): How to determine p in the Eq. ?

  **Reply:** Thanks for the suggestions. P stands for the number of periodic components. This can be determined by the reality. Such as considering 10 years periodic, 5 yeasr periodic, 2 years periodic, and 1 year periodic, then p equals to 4.

- Section 4: How about resolution?

  **Reply:** Thanks for the suggestions. Due to the complexity of fusion of multi-altimeter, it is hard to give a spatial resolution for the data set. The high resolution in title means high spatial coverage and large numbers of lakes.

- Table 3: RMS of Zhari Namco is 0.25m. But RMS is about 10.1cm in Sun et al. (Detecting lake level change from 1992 to 2019 of Zhari Namco in Tibet using altimetry data of TOPEX/Poseidon and Jason-1/2/3 missions. Frontiers in Earth Science, 2021, 9:640553, https://doi.org/10.3389/feart.2021.640553). Wang et al. (Robust, long-term lake level change from multiple satellite altimeters in Tibet: observing the rapid rise of Ngangzi Co over a new wetland. Remote Sensing, 11: 558, doi: 10.3390/rs11050558.) also shown the more precise lake levels.

  **Reply:** Thanks for the suggestions. For the RMSE of Zhari Namco, there is two possible reasons for this problem, one is that Zhari Namco is available on the eight altimeters in this manuscript, the data quality of Envisat is not stable, another is that fusion many different altimeter will also introduce noise. Actually, if we just consider Cryosat-2, Sentinel-3, Jason-2, and Jason-3, the RMSE is just 9.9 cm (as reported in Chen and Liao, 2020).

For the Ngangzi Co, no in situ gauge data are available in Ngangzi Co, so they are considering inter-compare with ICEsat and SARAL. In this manuscript, we just consider to compare with in-situ dada or existing product.

- Section 5: How about the physical mechanism? Why not the applications to climate change?

  **Reply:** There are many factors influencing lake level changes on the Tibetan Plateau, including Precipitation, and glacier and snow melt and degradation of permafrost brought by temperature changes, etc. This manuscript, as a dataset paper, mainly shows the changes of lake level and provides the basic data of lake level changes, so the physical mechanisms of lake level changes are not explored here, which will be further explored in future work.

  Lake level changes on the TP are an indicator of climate change in the region, which has been addressed in many papers (such as Gao et al., 2013; Hwang et al., 2016; 2019; Jiang et al., 2017), but here this manuscript does not expand on the application, but only points out the importance in the introduction.

**RC4**: ['Comment on essd-2022-313'](), Anonymous Referee #3, 28 Apr 2023

> The manuscript from Chen et al. presents the development and validation of a water elevation time series database for 362 lakes in the Qinghai-Tibetan Plateau from satellite multi-mission altimeters.
>
> General comment:
>
> The manuscript is easy to read and the methodology to derive water elevation is adapted, even if more information is needed (see specific comments below). However, I have the following main concerns:
>
> - The database is validated only for 8 lakes over 362. No information on the area of these lakes are provided, nor their locations, nor their hydrological regime. So, it's not easy to know if they are representative of many lakes within the 362 lakes in the database.

**Reply:** Thanks for the suggestions, the area, and locations of the lakes are included in the dataset. We will add an appendix table for including all the basic information (locations, areas, and also the hydrological regime info.

- For the remaining 354 lakes, no regional consistency analysis is done between neighbor lakes. It could be a way of cross-validating the database. Similarly, for each lake, no consistency between time series from different missions is done. It would also be a way to detect if some time series might be erroneous or not compared to measurements from other missions. The intermission bias is a good way also to check if the different missions are observing the same target (if one mission provide water elevations multiples decameters above/below other missions, then they are not observing the same target).

**Reply:** Thanks for the suggestions, the regional consistency analysis is a very useful tool for this dataset, at least we don't have so many gauge stations to evaluate, so we added the cross comparison with DAHITI (46 lakes), Hydroweb (40 lakes), G-REALM (8 lakes),and in Section 5.1, we added the trends analysis for the changes in the water levels of the lakes in different basins of the TP. This also a regional consistency analysis between neighbor lakes. At the same time, we don't think this could check whether the different missions are observing the same target. There exists a system bias between different missions.

- The comparison with other altimetry database is pretty weak. Lake trend is compared between the proposed database and the Hydroweb database. Why not comparing directly water elevations? Other altimetry databases (like DAHITI and G-REALM) should also be considered.

**Reply:** Thanks for the suggestions, we added the comparison with DAHITI (46 lakes), Hydroweb (40 lakes), G-REALM (8 lakes), and also direct with water elevation changes. Our initial purpose in comparing lake trends is because of various products have different errors. But the interannual trends obtained from these products should exhibit consistency. From your suggestion, we will also add the comparison with water elevation changes.

- Multiple satellite missions are used, which have different type of sensors. For example, ICESat is a lidar altimeter, which is quite different from nadir radar altimeters (both in vertical accuracy, sensitivity to clouds···). It is never explained, nor discussed. Pros and cons from lidar and nadir radar altimeters should be provided. For example, I would expect more accurate, but less measurements in time, from ICESat than from nadir radar altimeters.

**Reply:** Thanks for the suggestions. The different correction for ICEsat and nadir radar altimetry is in 3.1 section, ICEsat should consider the saturation correction, and radar altimetry should consider retracking correction. In this paper, our main goal is to generate a dataset, the different between lidar and nadir radar is not an important part.

> - Besides, there are some errors from nadir radar altimeters that are not discussed, but could have a huge impact on the database. This type of altimeters, in closed-loop tracking mode or with erroneous onboard DEM value in open-loop tracking mode, could lock their tracking window on the top of the surrounding topography near the lakes. If this topography is quite high compared to the lake (>tracking window size), the waveform will not sample the lake surface elevation. So, no matter how efficient is the retracking algorithm, it is not possible to retrieve the lake surface elevation. This point should be discussed in the manuscript. In addition to this type of error, when the lake is frozen, nadir radar altimeter could provide erroneous data. How is it delat with in the database?

**Reply:** Thanks for the suggestions. The OLTC was also considered in this study when we try to retrack the waveform. Due to the influence OLTC, the waveform will show without any peaks (just like the noise signal), this could be distinguished by using the data quality flag and waveform classification. When the lake is frozen, this will be processed using retracking, actually AMPDR has the ability to obtain the water level in winter which has already be discussed in Chen et al., 2021.

> - There are more and more published papers using satellite lidar data (ICESat, ICESat-2 and GEDI), see for example Luo et al. (2021) who studied 221 lakes in the QTP. How does your database compare to these studies?

**Reply:** Thanks for the suggestions. Comparing with other published papers sound like a good situation, but we suggest to compare with Legos and Dahiti dataset, Luo et al. (2021) has a gap losing 7 years. And our dataset is mainly using nadir radar altimetry.

> - Very few lake water elevation time series are presented (only three time series are shown, respectively in Figure 2, 3 and 4). They have rather a poor temporal sampling (figure 4) or strange elevation dynamic (figure 2).

**Reply:** Thanks for the suggestions. This should be the figure looks not good, we have already revised the figure to make it looks good. About the temporal sampling, this is the normal situation because some small lakes can only be monitored by the geodetic mission

Cryosat-2 without enough points, but we still add this inside, because these small lakes are also very useful for analyzing, which has been proven in Chen and Liao (2020).

- At the database repository, it is written that '196 lakes have the series from 2002 to 2021 in the 02-10 folder and 168 lakes have the series from 2010 to 2021 in the 10-21 folder'. This information should also be provided in the manuscript.

**Reply:** Thanks for the suggestions. This has already been included in the revised version.

- In the title, I don't understand why the term "high-resolution" is used. I would suggest to remove it.

**Reply:** Thanks for the suggestions. We will delete the high-resolution in title.

Specific comments:

There is an issue concerning the way ICESat-1 is presented in the text (firstly mentioned in the text at p.2 l.48). It is never mentioned that it is a lidar altimeter and not a nadir radar altimeter, like other satellite missions mentioned in the manuscript.

**Reply:** Thanks for the suggestions. The related information of ICEsat-1 has already been added in the revised version.

p.2 l.61-62: why not adding Icesat-2, as it covers the last part of the studied time span (2002-2021)?

**Reply:** Our aim was to generate a long time series dataset, and Icesat-2 was not considered since the time period covered by Icesat-2 was also covered by Sentinel-3. We will increase the use of Icesat-2 in subsequent studies.

p.3 l.83-84: Jason-1/2/3 were not only CNES mission, please edit (see for example https://www.eoportal.org/satellite-missions/jason-1, https://www.eoportal.org/satellite-missions/jason-2, and https://www.eoportal.org/satellite-missions/jason-3). These three missions have the same orbit, so why they do not observe to the number of lakes ? Besides, Jason-1 is well known for not providing much data over continents (e.g. see at the end of section 14.2.1 in Cretaux et al., 2017). Why do you have more data from Jason-1 than from Jason-3? I am very surprised by the

low number of lakes observed with Jason-3 (and the high number observed with Jason-1).

**Reply:** Thanks for the suggestions. Jason-1 will not provide much useful data in inland part, but this should be compared with Jason-2, because Jason-1 and Jason-2 all experience an interleaved orbit (Jason-2 from Oct. 2016 to June 2017, Jason-1 after February 2009), increasing the number of observed lakes. While Jason-3 does not experience an interleaved orbit during 2016-2021.

Reference:

Cretaux, J.-F., K. Nielsen, F. Frappart, F. Papa, S. Calmant, J. Benveniste (2017). Hydrological applications of satellite altimetry: rivers, lakes, man-made reservoirs, inundated areas. In: Stammer, D., Cazenave, A. (Eds.), Satellite Altimetry Over Oceans and Land Surfaces, Earth Observation of Global Changes. CRC Press, 2017.

Table 1 lacks some important information or have dubious information:

- You should add a column with the orbit repeat cycle for each mission (i.e. time sampling).

- The column with the duration is misleading. For example, if Envisat mission lasts from 2002 to 2012, in October 2010, its orbit changed (see https://www.aviso.altimetry.fr/en/missions/missions-passees/envisat.html) and therefore did its ground tracks and repeat cycle. The same goes for most missions cited in the table. Such information should be provided in the table or in the text. Besides, you should provide the time span over which you used the data. For example, for Envisat, you could only have used data from 2002 to 2010.

- I have some doubts concerning the diameter footprint provided in the last column of the table. For example, the AltiKa antenna footprint on the ground could be considered to be ~4 km (considering it corresponds to the 3-dB aperture angle of 0.6°) according to Steunou et al. (2015). Footprints of Envisat, and Jason-1/2/3 altimeters, because of Ku-band used is even coarser. Please edit the table and provide the references you used to derive information provided in Table 1.

**Reply:** Thanks for the suggestions. The orbit repeat cycle for each mission has already been added to the table. For the duration of the mission, we revised it according to different orbit, such as Jason-1/2/3 reference orbit, Jason-1/2 interleaved orbit (2009.02-2012.03 for Jason1, 2016.10-2017.05 for Jason2), Envisat reference orbit (2002.05-

2010.10), and Envisat extension orbit (2010.10-2012.04). All the antenna footprint has already been revised.

Reference:

Steunou N., J.-D. Desjonqueres, N. Picot, P. Sengenes, J. Noubel, and J.C. Poisson (2015). AltiKa altimeter: instrument description and in flight performance. Mar. Geodesy 38 (sup1), 22–42. http://dx.doi.org/10.1080/01490419.2014.988835

Table 2: What does the column 'Reference' and 'Mode' mean and correspond to? You should add a column with the yearly mean area (or something similar) of each lake.

**Reply:** Thanks for the suggestions. Reference means the geoid reference, Mode corresponds to the water level is a water level or water level anomaly. We have added the table footnotes to make it easier to follow. The yearly mean area we will add all lakes' area in the appendix table.

You should provide a map with lakes sampled with altimeters presented in section 2.2.1 and validation in situ data presented in section 2.2.2.

**Reply:** Thanks for the suggestions. The figure has already drawn with the location of the in situ data using the star symbol, and also the altimeters overpass orbit are also included, the satellite image map is regarded as the background.

Section 3.1: What is the 'satellite centroid correction'? Why do you need to consider 'ocean tide corrections' for lakes that are among the highest on Earth?

**Reply:** Thanks for the suggestions. The 'satellite centroid correction' has already been removed, because this correction has already been added in L1b processing chain. Actually, the ocean tide corrections are all zero in Tibetan Plateau lakes, we mentioned it here is for making this formula more complete.

p.4 lines 110-120 are almost a copy/paste of Chen and Liao (2020)

**Reply:** Thank for your suggestions. We have modified this part.

p.4 l.122 and following references to Chen et al. (2020): Chen et al. (2020) is not provided in the References section.

**Reply:** Thanks for the suggestions. This reference has been added.

Chen, J., Liao, J., Wang, C. (2021). Improved lake level estimation from radar altimeter using an automatic multiscale-based peak detection retracker[J]. IEEE Journal of Selected Topics in Applied Earth Observations and Remote Sensing, 14: 1246-1259. doi.org/10.1109/JSTARS.2021.3035686.

> Section 3.1.1 assumes readers are familiar with the AMPDR retracker, which is not a commonly used retrakcer. So, more information on this retracker is needed and you should define the acronyms and variables mentioned (like HDEM, DistanceThresh⋯). As written in lines 124 to 126, it seems that you don't have any bias using AMPDR with Jason-2/3, Sentinel-3A/B and Cryosat-2, contrarily to other missions. I have observed bias between J2/3 and S3/B over many lakes and rivers using OCOG retracker. I am therefore more surprised that you don't have bias with these missions, rather than you observe bias with other missions (and according to section 3.2, it seems that you have biases between each consecutive mission).

**Reply:** Thanks for the suggestions. Actually, this is a dataset description, so we don't include too many retracking methods, as it was already published. But we also added the acronyms and variables mentioned to support this method can be easy to follow.

AMPDR is the retracking suitable for Jason-2/3, Sentinel-3A/B and Cryosat-2, both these four satellites can get good results, which has been proved in Chen et al. (2021). This is related to the OLTC and SAR technique but still has some observations not good resulting in the bias you see, this will be finally removed from the processing of merging multi-altimetry. But for the other missions, the data quality is not so much good in inland, so we make some improve for AMPDR to make the statistics threshold more reasonable.

> p.6 l.162: How the training set of 300 waveforms have been selected? What type of lakes do they cover?

**Reply:** Thanks for the suggestions. Waveforms are uniformly selected with various types of lakes, and also various locations of the lakes. Not for some type of lakes.

> p.6 l.167: Why excluding tracks with fewer than 5 observations?

**Reply:** Thanks for the suggestions. This is coming from the AMPDR, this retracker are considering the statistics of the along-track water level, fewer than 5 observations will not be enough for giving stable statistics and also mean the observation is easier be polluted at terrain signals.

> Section 3.1.3, I don't see what is the purpose of using tsHydro (equations 2 and 3). You should explain in more details what is the purpose of this tool, what it is supposed to do, why it is needed and all the parameters used

(sigmaRW is not explained for example), why specifically equations 2 and 3 have been selected (why this model has been chosen). Providing this information in the manuscript is important, even if they are present in Nielsen et al. (2015).

**Reply:** Thanks for the suggestions. We have added the explanation for the tshydro, such as the dynamic model formulation and its explanation.

p.7 l.182: I guess the 'reference plane' is the geoid or ellipsoid used to reference water. It would be good to provide explicitly the definition in the text.

**Reply:** Thanks for the suggestions. Yes, the 'reference plane' is geoid, we have added it in revised version.

Section 4.2 and Figure 6: Why comparing trend and not directly water elevation time series? I would prefer to see a comparison between time series. There are other altimetry water elevation databases like DAHITI (https://dahiti.dgfi.tum.de) or G-REALM (https://ipad.fas.usda.gov/cropexplorer/global_reservoir/ and https://blueice.gsfc.nasa.gov/gwm/lake/Index). You should also add these databases in your comparison

**Reply:** Thanks for the suggestions, we will add the comparison with DAHITI (46 lakes), G-REALM (8 lakes), and also direct with water elevation changes. Our initial purpose in comparing lake trends is because of various products have different errors. But the interannual trends obtained from these products should exhibit consistency. From your suggestion, we will also include the comparison with water elevation changes.

Section 5.2 is too qualitative, some assertions are not really supported by the figure (discharge seems to raise before the lake level, which is not coherent with the fact that discharge is regulated by the lakes; no information on the used precipitation is provided; there is some connection between precipitation and lake level variation and discharge; discharge time series might not be fully validated, given its 'stair steps' shape over some periods) and does not provide anything to your database. I suggest to delete this section.

**Reply:** Section 5.2 focuses on a case of lake level application, the purpose of which is to show that lake level changes can reflect river regulation. Without the influence of rainfall,

the water level changes of the two lakes are consistent with the discharge changes along the Yellow River, while the downstream rivers, which are not subject to the regulation of the lakes, have uneven discharge changes. Since the discharge data from gauge stations were applied, mainly qualitative relationships were derived in this study.

> p.15. l.320: Why does your database could be labeled as 'high-resolution datasets'?

**Reply:** Thanks for the suggestions, the high-resolution datasets is for spatial, but maybe not unclear here, we will delete it.

> I download the full database and plotted all time series. Some time series looks really good, but some other raise some questions. I have not been able to load the plot on the server, but the following time series illustrates some of my observations:
>
> - Buergacuo_Lake_Water Level.txt has a clear different behavior before and after 2011. Is it realistic?

**Reply:** The lake was originally recharged by a spring. It can only be inferred that the water level tends to rise steadily after 2011 because the groundwater recharge from glacial melt is greater than the spring recharge after the temperature rises.

> - Co_Ngoin1_Water Level.txt has a clear annual cycle after 2011, which is not the case before and it seems strange to me.

**Reply:** Thanks for the suggestions, before 2011, it is very hard to see this situation because only ICEsat can observe it. But after 2011, Cryosat-2 and Sentinel-3 could.

> - Kusai_Lake_Water Level.txt has an 8m water elevation increase in less than a month. Could it be due to an intermission bias?

**Reply:** Thanks for the suggestions, this is not the intermission bias. Kusai Lake experienced an abrupt expansion in 2011, resulting from the dike break of an upstream lake, named Lake Zhuonai. Li et al. 2019 also reported this situation.

> - Laorite_Co_Water Level.txt, for some time series beofre 2011 when there are few points, time series seems to have a smooth curvy shape, which is not the case for time period with more data. Could it be due to a smoothing form the tsHydro processing?

**Reply:** Thanks for the suggestions, but we think this is not the case, before 2011 looks smooth mainly because the tempoal resolution is not enough for ICEsat overpassing this lake.

- Gyesar_Co_Water Level.txt has too few measurements, is it worthwhile to provide this time series and compute trend?

**Reply:** Thanks for the suggestions, although it has a few points, but still can be very useful for estimating the lake level annual change rate.

- Xiaoquan_Lake_Water Level.txt has a 80m increase for few time steps, which does not seem realistic

**Reply:** Thanks for the suggestions, this lake has some problems when generating time series, some big bias observations was considering, we will consider remove this lake.

---

## Author Response (AR2)

**Responses to the comments on the Manuscript "A dataset for lake level changes in the Tibetan Plateau from 2002 or 2010 to 2021 using multi-altimeter data"**

Dear editor,

The authors would like to express thanks to the anonymous reviewers for their voluntary work and the constructive comments to improve this manuscript. All of the comments are of great benefit to us. During the past few days, we did much work to revise the manuscript according to the reviewer's comments. All of the comments have been addressed. Our revisions are as follows.

07 Oct 2024

**Topic editor decision: Reconsider after major revisions**

by Birgit Heim

**Public justification (visible to the public if the article is accepted and published)**:

Dear Authors and Colleagues

Thank you for your contributions. Many thanks for the revisions and thanks for the authors for the replies and the edits in your manuscript.

The manuscript, the data description and data publication do not yet fulfill the requirements of ESSD and improvements are needed in form of a major revision of the manuscript and a minor revision of the dataset publication.

Please consider the reviewers comments and technically discuss all issues, specifically concerning the comments of reviewer 3. Please also consider the critical comments from the open discussion.

Minor revision of the PANGAEA data publication: please provide a technical read me, e.g. for users of the downloaded lake level data in the current format it is not easy to understand that the two subfolders represent two different time slices, you could e.g. attach a detailed technical product guide (e.g. also including figures and tables) in pdf format. Attention: in your data files the symbols for the unit of Water level, uncertainty (WL unc) seem to be corrupt. Could you please check and correct?

After reading the reviewers comments and concerns, we also suggest to add a quality information to each lake level data product.

***Reply:***

Thank you for the suggestion. In the last month, we have carefully revised our manuscript in response to the reviewers' suggestions. The dataset format has been updated: we consolidated all lake time series data into a single archive, dataset.zip, organized as 361 individual entities named by lake. The previous subfolder structure has been removed for clarity.

Additionally, we included two technical readme files to help data interpretation: readme_dataset_info.kml and readme_dataset_info.md. The .kml file can be opened directly in Google Earth Pro, providing an intuitive geographic view of the dataset, while the .md file lists detailed information for each lake in the title of each corresponding file, offering a detail reference to all included data.

Report #1

**Suggestions for revision or reasons for rejection**

(visible to the public if the article is accepted and published)

The authors have adressed all comments and proposals. The manuscript has been revised. I recommend the manuscript should be accpeted for publication in ESSD.

***Reply:***

Thank you for the suggestion.

**Report #2**

Submitted on 16 Sep 2024
Anonymous referee #1

**Anonymous during peer-review:** **Yes** No

**Anonymous in acknowledgements of published article:** **Yes** No

**Checklist for reviewers**

| 1) Originality | Excellent Good Fair **Poor** |
|---|---|
| 2) Significance | |
| Uniqueness | Excellent Good **Fair** Poor |
| Usefulness | Excellent Good **Fair** Poor |
| Completeness | Excellent Good **Fair** Poor |
| 3) Presentation quality | Excellent Good Fair **Poor** |
| 4) Data quality | Excellent Good **Fair** Poor |

**For final publication, the manuscript should be**

accepted as is

accepted subject to **technical corrections**

accepted subject to **minor revisions** (review by editor)

reconsidered after **major revisions**

rejected

**Were a revised manuscript to be sent for another round of reviews:**

**I would be willing to review the revised manuscript.**

I would not be willing to review the revised manuscript.

**Suggestions for revision or reasons for rejection**
(visible to the public if the article is accepted and published)

The authors have improved the manuscript. However, there are still some obvious typographical errors. These indicate that the authors did not read the manuscript carefully. In addition, this lake level dataset is not new now.

Main comments:
1) Why was ICESat-2 not considered?
***Reply:***

Thank you for the suggestion. The Icesat-2 does not have enough temporal resolution (91 days), the time period covered by Icesat-2 was also covered by Sentinel-3. Additionally, we are mainly consider radar altimeter data here.

2) Table 1: Please add some new studies for comparison, especially after 2019.
***Reply:***

Thank you for the suggestion. We added Luo et al. (2021) in Table 1.

3) Figures 2 and 5, can these two flowcharts be combined?
***Reply:***

Thank you for the suggestion. Sorry for this, it would be hard to combine together. Figure 2 is only for retracking, while figure 5 is to merge the multi-satellite time-series.

Specific comments:
L10: "Tibet Plateau (TP)" to "Tibetan Plateau (TP)"
***Reply:***

Thank you for the suggestion. It has been revised.

L50: "Tibetan Plateau" to "TP", and other corrections are similar.
***Reply:***

Thank you for the suggestion. It has been revised, and also other similar place.

Table 2: How about Sentinel-6MF? See reference: doi: 10.1016/j.asr.2024.04.006
***Reply:***

Thank you for the suggestion. We do not use Sentinel-6 here, the dataset here only until 2021/07. Sentinel-6 only have no more than 1 year's data, so we donnot consider it here.

L160: ICE-1 is ICESat-1?

*Reply:*

Thank you for the suggestion. Actually not, it is OCOG retracker, we change it already in revised version.

*Reply:*

Thank you for the suggestion. It has been revised.

Figure 3: longitude to Longitude

*Reply:*

Thank you for the suggestion. It has been revised and improved.

Figure 4: Some texts are too small, and Water Level to Water level

*Reply:*

Thank you for the suggestion. It has been revised.

L290: 6364km2 to 6364 km2

*Reply:*

Thank you for the suggestion. It has been revised.

Figure 10: area is Level?

*Reply:*

Thank you for the suggestion. This is not for mentioned Level, we are describing the relative proportions based on the lake areas in the basin.

Figures: All figures need to improve, some texts are too large, and some are too small.

*Reply:*

Thank you for the suggestion. All the figures has been remade and impoved.

**Report #3**

Submitted on 04 Oct 2024
Anonymous referee #3

**Anonymous during peer-review:**                    **Yes** No
**Anonymous in acknowledgements of published article:** **Yes** No

**Checklist for reviewers**

| 1) Originality | Excellent Good **Fair** Poor |
| --- | --- |

**2) Significance**

| | |
|---|---|
| **Uniqueness** | Excellent Good **Fair** Poor |
| **Usefulness** | Excellent **Good** Fair Poor |
| **Completeness** | Excellent **Good** Fair Poor |
| **3) Presentation quality** | Excellent **Good** Fair Poor |
| **4) Data quality** | Excellent **Good** Fair Poor |

**For final publication, the manuscript should be**

accepted as is

accepted subject to **technical corrections**

accepted subject to **minor revisions** (review by editor)

**reconsidered after major revisions**

**rejected**

**Were a revised manuscript to be sent for another round of reviews:**

**I would be willing to review the revised manuscript.**

I would not be willing to review the revised manuscript.

**Suggestions for revision or reasons for rejection**
(visible to the public if the article is accepted and published)

The revised version of the manuscript from Chen et al. submitted to ESSD has been quite improved and replied to some of my main comment. The addition of Table 1 comparing this study to ones previously published and the comparison of their new database with the ones from Dahiti, G-REALM and Hydroweb provides more insight in the novelty and robustness of this new database. I think that this manuscript and the associated database has the potential to be published in ESSD. However, I still think it is needed to improve the manuscript considering the following general comments:

- Even if it has been clarified in the abstract, I am still bothered that the title is 'A dataset for lake level changes in the Tibetan Plateau from 2002 to 2021 using multi-altimeter data'. 46 % of the lakes in the database (167 out of 361) have a time series covering the 2010-2021 time span and not the 2002-2021 time span. I would prefer to have a title like: 'A dataset for lake level changes in the Tibetan Plateau from 2002 or 2010 to 2021 using multi-altimeter data'.

*Reply:*

Thank you for the suggestion. We change the title into 'A dataset for lake level changes in the Tibetan Plateau from 2002 or 2010 to 2021 using multi-altimeter data'.

- The text needs to be clarified, as some sentences are difficult to understand, especially in sections 3.1.1, 3.1.2 and 3.1.3.

*Reply:*

Thank you for the suggestion. We improve the text and make it easier to understand and following. See more details in revised version.

- Section 5.1 presents trends computed over 2002-2021, but given that almost half of the lakes cover only 2010-2021, it seems to me that half of the trends computed and shown on the maps are not computed on the same period. This part should be revised (for more details, see one of my comment below).
**Reply:**

Thank you for the suggestion. We revised the text, figures, and tables in section 5.1, by only considering the analyze between 2010–2021-time span. See the details in revised version.

- I am still not convinced about section 5.2 ('Exploring the responses of the lake levels to river regulation'), which is quite qualitative and some assertions made are assumption that would require much more work to be validated. I think this section could be safely removed.
**Reply:**

Thank you for the suggestion. We remove this section in revised version.

Specific comments:

In the reply to my previous following comment: 'The intermission bias is a good way also to check if the different missions are observing the same target (if one mission provide water elevations multiples decametres above/below other missions, then they are not observing the same target).' Authors' reply concerning the bias is:
'we don't think this could check whether the different missions are observing the same target. There exists a system bias between different missions.'
The authors missed my point. I know that there is a bias between different mission. This bias is dependent of the retracker used, the technical characteristics of the altimeter, the way the waveforms are selected and if multiple waveforms are retracked if the median WSE is computed or not. However, this bias should be between few cm and few meters. If the bias is >10m, then the altimeter is not observing the same targets, such bias could not be explained with the sources of bias I listed previously. It could be specifically the case between missions in closed-loop tracking mode and the ones in open-loop tracking mode.
**Reply:**

Thank you for the comment. We understand that a significant bias (e.g., >10 meters) likely indicates that different targets are being observed, which cannot be fully explained by typical system or retracker biases.

In our previous response, we focused on known biases that can arise from differences in retracking methods, altimeter characteristics, and waveform selection, which we expect to be within a range of a few centimeters to a few meters. However, we agree that a bias exceeding this range could indeed indicate that the altimeter is observing a different target, particularly when comparing closed-loop and open-loop tracking modes. Actually, when we doing the fusion of multi-altimeter time series, the intermission bias analyze is also been used, since we are going to only merge the time series when $\overline{Series1_{ref}} - \overline{Series2_{ini}} < 10$ , for those have too much bias we believe it comes from noise or other

bright target. We also added the info in Section 3.2.

In Section 4.3, we have now added a discussion on the specific challenges related to large biases in inter-mission comparisons and the possibility of such cases indicating different observed targets.

Authors' comment to one of my previous comment stated: 'In this paper, our main goal is to generate a dataset, the different between lidar and nadir radar is not an important part'
I don't agree with this statement. Such sensor differences will make the time series accuracy quite heterogeneous in time and could help to explain why some portion(s) of the time series is(are) noisier than others. That's why in section 2.2.1 ('Multi-altimeter data'), it should be clearly stated that ICESat-1 is a lidar altimeter, contrarily to other altimeters, that are nadir radar altimeters.
**Reply:**

Thank you for the comment. Yes, I agree with you. Different sensor could make temporal resolution heterogeneous in different period. We improve the sentence in section 2.2.1, to better describe the different between ICEsat used here and other radar altimeters.

Authors stated: 'Due to the influence OLTC, the waveform will show without any peaks (just like the noise signal), this could be distinguished by using the data quality flag and waveform classification'
I do not fully agree with this statement. If the tracking window is not observing any continental surface, then I agree that only noise is recorded and this case is easy to handle. However, sometime the tracking window can see another lake or river and therefore have a valid waveform, but it will not observe the wanted lake. How such case is handled with your processing chain? It is a source of errors in your database that might happen for some lakes. More generally, it is needed to include (for example in section 3.2) a paragraph on potential sources of errors or inconsistency in the time series in the database.
**Reply:**

Thank you for the comment. I agree that, sometimes, the tracking window may primarily capture signals from other rivers or lakes. However, the waveform range covers approximately 60 meters (0.4684 * 128), meaning that if the height difference between the target lake and nearby bright features is less than 60 meters, we can still obtain accurate information, even if the OLTC DEM is not entirely correct. Additionally, to minimize the chances of capturing incorrect lake signals, we use DEM to select height values, exporting only lake levels that fall within the range of $H_{DEM} \pm 20$ m. However, this approach is not fully correct, especially in regions where neighboring water bodies are within similar elevation ranges. Such cases could introduce inconsistencies in the time series for certain lakes, particularly where OLTC DEM values have changed over time, affecting the tracking window's focus.

But the case you mentioned, I also see once in Swiss Lake. For example, Mattenalpsee has an elevation of approximately 1800 meters, while Oberaarsee is around 2300 meters—a difference of over 500 meters. When using the same track observations, due to changes in OLTC settings, the OLTC DEM value was set to 2300 meters before 2020, making it possible to monitor Oberaarsee. However, after 2020, the OLTC DEM value was adjusted to 1800 meters, enabling monitoring of Mattenalpsee instead.

We have added a new section (Section 4.3) on potential sources of error to detail possible issues, providing clearer information for readers and dataset users.

Authors' replied to one of my comment: 'Actually, the ocean tide corrections are all zero in Tibetan Plateau lakes, we mentioned it here is for making this formula more complete.'
I suggest to remove the reference to ocean tide correction or state explicitly in the text it is equal to 0, otherwise it seems strange to have this type of correction on the TP.
***Reply:***
>Thank you for the suggestion. We remove ocean tide correction in revised version.

Table 1: Databases from Li et al. (2019) and Zhang et al. (2019) are also public (like Dahiti, G-REALM and Hydroweb), so why did not you also compare your database to this two in section 4.2?
In my previous review, I suggested to reference the database from Luo et al. (2021) who studied 221 lakes in the TP, why did not the authors consider this article and database? I think you should include it.
Reference:
Luo S., C. Song, P. Zhan, K. Liu, T. Chen, W. Li, and L. Ke (2021). Refined estimation of lake water level and storage changes on the Tibetan Plateau from ICESat/ICESat-2. Catena, 200, 105177, https://doi.org/10.1016/j.catena.2021.105177
***Reply:***
>Thank you for the suggestion. We added the comparison between our database and Li et al. (2019) in section 4.2 and Table A1. But we realize Zhang et al. (2019) is actually not available now, so we cannot do the comparison here, to make it clear, we change the table 1, Zhang et al. (2019) from public into not public. Also, Luo et al. (2021) has been added in Table 1.

Table 2, diameter of footprint for Envisat, Jason-1/2/3 and Saral are still wrong, see Steunou et al. (2015):
'the 3-dB footprint radius is about 4 km versus 15 km for Poseidon on Jason missions'
So, the diameter footprint for Jason-1/2/3 should 30 km and diameter footprint for Saral should be 8 km. Envisat radar altimeter diameter footprint should be similar to the ones for Jason, but I let the authors do their research for the value. For Sentinel-3A and Crysoat-2, where does the along and across track footprint diameter come from? According to https://sentiwiki.copernicus.eu/web/s3-altimetry-instruments, Sentinbel-3A/B radar altimeter 'Radar footprint diameter (LRM/P-LRM)' is ~15km, whereas 'Along-track footprint (SAR mode)' is ~330m. Please edit Table 2 'Diameter of footprint (km)' accordingly.
Reference:
Steunou N., J.-D. Desjonqueres, N. Picot, P. Sengenes, J. Noubel, and J.C. Poisson (2015). AltiKa altimeter: instrument description and in flight performance. Mar. Geodesy 38 (sup1), 22–42. http://dx.doi.org/10.1080/01490419.2014.988835
>***Reply:***
>>Thank you for the suggestion.
>
>>LRM/P-LRM footprint diameter is $D = 2h * \tan(\frac{\theta_{3db}}{2})$
>
>>SAR/SARin footprint width across-track direction: $D = 2 * \sqrt{h \cdot \frac{c}{B}}$

SAR/SARin footprint width along-track direction: $\Delta x = h \frac{\lambda}{2 \cdot N \cdot v} PRF$

Envisat antenna beamwidth at -3 db is around 1.5 °, resulting $D_{Envisat} \approx 20\ km$.

Cryosat-2 $\lambda = 0.021m,\ PRF = 18181Hz$,

   resulting $D_{Cryosat2} \approx 1.65km,\ \Delta x_{Cryosat2} = 0.3km$

Sentinel-3 $\lambda = 0.0221m,\ PRF = 17825Hz$,

   resulting $D_{Sentinel3} \approx 1.75km,\ \Delta x_{Sentinel3} = 0.33km$

To distinguish the different between SAR and LRM/P-LRM. We added a table notes in revised version. "the footprint for SAR/SARin can be approximated by a rectangle given with the footprint width in across track and along track". And change the Diameter of footprint into width of footprint.
Reference:

1. https://earth.esa.int/eogateway/documents/20142/37627/CryoSat-Footprints-ESA-Aresys.pdf
2. https://earth.esa.int/eogateway/documents/20142/37627/ENVISAT-RA-2-MWR-Product-Handbook.pdf
3. https://sentinel.esa.int/documents/247904/4871083/Sentinel-3+SRAL+Land+User+Handbook+V1.1.pdf

l.135: 'the implementation of retracking correction', why do you write 'retracking correction'? It should be the retracking of the waveform, rather than a correction, should not it?

**Reply:**

  Thank you for the suggestion. We revised it in revised version.

l.134: 'due to the potential interference or submergence of waveforms by signals from adjacent land areas' → this sentence is not clear, please rephrase.

**Reply:**

  Thank you for the suggestion. We rephrased it into "due to the potential interference or submergence of water signals by those from adjacent land areas".

l.138: 'can get good results (Chen and Liao, 2020)' → 'good results' is quite qualitative. You should provide some metrics from Chen and Liao (2020) to provide more quantitative information to this statement.

**Reply:**

  Thank you for the suggestion. We revised this sentence into: "In this study, we employed the automatic multiscale-based peak detection retracker (AMPDR) (Chen et al., 2021). The Jason-2/3, Sentinel-3A/B, and CryoSat-2 satellites are all suitable for providing precise measurements, with average accuracies of 0.18 m, 0.14 m, and 0.15 m when compared to gauges, respectively."

Figure 2 is not always easy to read. For example, what does 'For each track, determine the multiscale-based retracking level' or 'Determine the abnormal track by the DEM and the water level derived from the neighbouring cycle' mean? Some line does not have arrow, so it's difficult to know how these boxes are used in the flowchart.

**Reply:**

Thank you for the suggestion. The figure 2 has been improved, and "For each track, determine the multiscale-based retracking level" revised into "For each track, determine the multiscale-based retracking level from sub-waveform using threshold retracker.", "Determine the abnormal track by the DEM and the water level derived from the neighbouring cycle" revised into "Determine the abnormal track by the range of DEM ±20m and the water level derived from the neighboring cycle". The details info can be seen in next text within "Following this, a second run of AMPDR was performed to retrack abnormal tracks identified by checking if the current cycle's water level fell within the range of the Digital Elevation Model (DEM) ± 20 m and by comparing it with water levels from neighboring cycles, particularly when a significant discrepancy or abrupt change was detected in current cycles."

l.155 to l.163: Description of the procedure (and Figure 2) is not clear and needs to be rewritten. Why are you using also ICE-1 retracker in addition to the AMPDR? The benefits of using AMPDR instead of Ice-1 should be provided. The use of the DEM is not clear to me either.

**Reply:**

Thank you for the suggestion. The ICE-1 retracker is not used directly for retracking, it is actually to provide a possible water level into construction the point cloud of AMPDR retracker, this is because the multiscale-based adaptive threshold value would fail in some situations, such as sometimes, the signal from LRM mode altimeter is very noise, the adaptive threshold value may find too many noise which make it hard to create a good point-cloud. We revised this sentence to make it clear. "Additionally, a retracking point from the OCOG algorithm was incorporated into the AMPDR to assist in constructing the "point cloud" and CDF. This integration addresses specific cases where AMPDR's adaptive thresholding may encounter challenges."

l.169-170: 'Due to the use of a 1 km buffer to pick out the shape of the available footprints, there would be many noise footprints caused by the reflected signals of the terrain or by the scatter signals of the off-nadir points' You should define what is a 'noise footprint'. I don't understand what you mean, as a footprint is the observed area on the ground, it's not noisy by itself. A waveform could be noisy, but you should also define what you mean by noise. Multiple targets could send back energy to the satellite, which will be recorded in the waveform. These cases are difficult to retrack. This section describes how waveforms are selected within the polygon of the lake, whereas some other that are more difficult to process are excluded. So, it seems to me this section should be more labelled 'waveforms selection' or something like that.

**Reply:**

Thank you for the suggestion. This sentence maybe not clear enough. We are not picking up the footprints, we are selecting the observations available on the shape (with 1 km buffer), this could increase the number of observation, but also introduce uncertainty, the waveform of some observations could be noisy (hard to find the leading edge), which could be caused by the signals of terrain or complex signals of off-nadir observations.

We revised this sentence into "By selecting observations within a 1 km buffer around the lake shape, we capture additional data points. However, this approach can also introduce uncertainty, as some observations may contain noisy waveforms that complicate the retracking process. This noise can result from signals reflected off surrounding terrain or from off-nadir observations."

The section has been revised into 'waveforms selection'.

In equation 2 and 3, the 'i' and 'j' indices and the variable 'z' must be defined.
 *Reply:*
Thank you for the suggestion. $t_i$ is the time of the $i$-th time step, j is the number of observations in given time, and $z_i$ is a random noise term following a standard normal distribution. The definition has been added in revised version.

l.193: 'According to the Laplace estimation, the mean value of the range was selected to represent the water level of the lake for each cycle' → What is the 'Laplace estimation'? Why does it imply to select the mean and not something else (for example the median)?
 *Reply:*
Thank you for the suggestion. According to the process model and observation model in Eq.2 and Eq.3. The likelihood function of the true water level is non-Gaussian, so the Laplace approximation is used in "tsHydro". More details can be seen in Nielsen et al. (2015).
But to make it clearer, we revised this sentence and remove Laplace estimation, which is not main point in the paper. The predictions of the true heights $\hat{H}^{(true)}$ is the estimate of the water level of the lake for each cycle.

As there are some lakes with time series starting in 2002 and some other starting in 2010, on which period have the trends provided in the text and figures in section 5.1 been computed? If it varies depending of the time series, then it might be difficult to compare trends not computed on the same period. I think figure 8 to 10 and Tables 5 and 6 should be computed over 2010-2021, and other figures and tables should be added only for lakes with time series really covering the 2002-2021 time span. Besides, the equation and method to compute the linear trend should be provided. It should also be added if the trends are statistically significant and you should only consider lakes with statistically significant trends. For the unit of the trend, why using 'm/a' and not, for example 'm/y' (i.e. for meter/year) which seems to be more explicit to me?
 *Reply:*
Thank you for the suggestion. We revised the text, figures, and tables in section 5.1, by emphasize considering the analyze between 2010–2021-time span, while for the period over 2002-2021, we added the summarize text, and also the Basin change info in table 5. See the details in revised version.
The equation and method to compute the linear trend was added in the beginning of Section 5.1.
The unit of the trend has been changed into 'm/y' in revised version.

Figure 8: I find it difficult to distinguish the different size of the arrows. It might be easier to have four different symbols for the four different ranges of trend.
 *Reply:*
Thank you for the suggestion. The figure 8 has been improved by changing the different symbol.

Table 5: Does the 'Annual rate of change' correspond to the mean of all trends within the

considered lake area range? For this annual rate and the mean rate of rise/decrease, is the lake trend weighted in some way by the lake size or is it the arithmetic mean?

*Reply:*

Thank you for the suggestion. It is actually weighted by lake size. To make it clear, we added the info in revised version.

l.384: Why are you mentioning a 'modified waveform retracker'? You used the published retracker from Chen et al. (2021), did not you? Did you modified the retracker compared to Chen et al. (2021)?

*Reply:*

Thank you for the suggestion. We are using AMPDR from Chen et al. (2021), but when processing with Jason-1, Envisat, and SARAL, AMPDR could retrack fail in some cycles, this time we are going to do the second run of retracking, this can be seen more details in section 3.1.1. But you are right, in conclusion, using 'modified waveform retracker' is not clear, we changed it into "two-step AMPDR retracker".

---

## Author Response (AR3)

**Responses to the comments on the Manuscript " A dataset for lake level changes in the Tibetan Plateau from 2002 or 2010 to 2021 using multi-altimeter data "**

Dear editor,

The authors would like to express thanks to the anonymous reviewers for their voluntary work and the constructive comments to improve this manuscript. All of the comments are of great benefit to us. During the past few days, we did much work to revise the manuscript according to the reviewer's comments. All of the comments have been addressed. Our revisions are as follows.

**Editor comments::**

Thank you very much for your new contributions. Many thanks to the reviewer for the deep-going reviews and thanks for the authors for the detailed replies and the edits on the manuscript and dataset on lake level time series on the Tibetan plateau. The manuscript is now in minor revision and we would like to thank the authors, who revised manuscript and dataset publication and our referees for these constructive efforts. Dear authors, please consider the reviewers comments and technically discuss all issues. Please also consider the critical final comment from the open discussion. There is also the editorial minor request for a minor addition to the PANGAEA data publication that will not change the status of the DOI-referenced data publication in 2024. Thank you for providing the detailed information per lake in form of the kml and the md files that is a very user-friendly format. As your data set is very complex please provide also this overview in form of a short technical product guide on the file structures (and optimally on the processing scheme also including figures and maps) in pdf format that could be attached by PANGAEA to the abstract text, i.e. this would not be part of the downloaded dataset but is part of the abstract text and could be downloaded from there.

*Reply:*

Thank you for your constructive feedback and the opportunity to improve our manuscript and dataset. We sincerely appreciate the reviewers' insightful comments and have carefully addressed all technical and editorial suggestions. As requested, we have prepared a Technical Product Guide (PDF) summarizing: File structures: Organization of TXT files, KML, and MD files. And region maps, processing workflow.

This guide has been submitted to PANGAEA for attachment to the abstract page (separate from the dataset download) to improve user accessibility, which will not alter the DOI-referenced data publication (2024 status).

**Reviewer1:**

Authors have satisfactorily addressed my main concerns and I thank them for their careful answers and edits to the manuscript. I think the paper could published in ESSD, provided the authors address my specific comments below. There is no need to have a new round of review on my side.

***Reply:***

Thank you for your positive feedback and for acknowledging our revisions. We sincerely appreciate the time and effort you have dedicated to reviewing our manuscript.

We have carefully addressed all of your specific comments (point-by-point responses are provided below).

Specific comments:

Abstract: 'The period for the lake level change series, which affords high accuracy, can be much longer for many lake systems.' → this sentence is not clear and the term 'high accuracy' is quite subjective. Here you should provide the RMSE (and correlation) when your database is compared to few available in situ gauges (and recall the number of gauges) and the cross-validation with Dahiti, Hydroweb.next and G-REALM. It will be more informative.

***Reply:***

Thank you for highlighting the need for clearer validation metrics. We have revised the abstract to incorporate specific accuracy statistics and cross-validation results. Below are the details. "The lake level change series shows good consistency with in situ measurements, demonstrating a median RMSE of 0.19 m across 8 validation gauges. The dataset further exhibits robust agreement with established satellite altimetry products (DAHITI, Hydroweb.next, and G-REALM), with median RMSE values below 0.30 m in all cross-validation comparisons."

In Table 1 the time period and number of lakes of each dataset might need to be updated. This is especially true for Dahiti and Hydroweb (which is now hydroweb.next), which have near real time time series for some lakes (I don't know if this is the case for lakes in the Tibetan Plateau) and new lakes might have been added to the database (here again, I do not know if this is the case for the TP). If it has changed, then the text in section 1 should be updated accordingly. In Table 1, you should also add G-REALM (https://ipad.fas.usda.gov/cropexplorer/global_reservoir/), especially as you are comparing your results with this database in section 4.2.

***Reply:***

Thank you for your suggestion, Hydroweb is updated the number of lakes in TP, now it has 46 lakes, Dahiti is not changing during the period. The G-REALM is also added in the table and text.

In section 2.2.1, 'ICESat-1 is a lidar altimeter, distinct from above radar altimeters. Its technique provides high spatial resolution and small footprint, but results in less measurements over time.' This sentence should be expanded with few words like: 'because of its orbit repeat period of 91 days and impact of clouds'. Otherwise, some people might think that less measurements over time are only due to the sensor itself.

***Reply:***

Thank you for your suggestion, we revised the sentence into "ICESat-1 is a lidar altimeter, distinct from the above radar altimeters. Its technique provides high spatial resolution and small footprint, but results in fewer measurements over time due to its 91-day orbit repeat period and frequent data gaps caused by cloud obstruction."

Section 3.1.1, 'potential interference or submergence of water signals by those from adjacent land areas' → I don't think 'submergence' is the correct term here. Would 'interference of adjacent land areas signals with signal from water body' fit authors' message?

***Reply:***

Thank you for your suggestion, we revised the sentence into "due to the interference of adjacent land areas signals with signal from water body."

The threshold level variable name is slightly different between Figure 2 (DistanceThres) and the text (e.g. DistanceThresh, p.6 lines 164 and 166). Please, use the same variable name in the figures and in the text.

***Reply:***

Thank you for your suggestion, we revised the figures to make sure the same variable name in figures and in the text.

Line 148 page 6, HDEM should be defined here (currently, it is explained in Figure 2 and in line 162 page 6)

***Reply:***

Thank you for your suggestion, we revised the sentence into "the optimal retracked levels should be within the range of the Digital Elevation Model (DEM)-based reference elevation $H\_DEM \pm 20$ m."

Figure 3 is a good figure to show the impact of the two-step retracker. In the legend, could you give the mean longitude and latitude of the measurements and the name of the lake to help reader to locate this lake? The water level time series provided in figure 3 has a 'zig-zag' shape for many cycles. Have you been able to validate this time series against in situ water level? If yes, it would be good to mention it in the text.

***Reply:***

  Thank you for your suggestion, we added the lake name and which satellite (Ayakkum Lake using SARAL measurement) been used here in figure title to make the reader locating this lake.

  Unfortunately, this lake does not have an insitu data, but by cross validate with other satellite in the same lake, the time series are fitting good with time series from different satellite. The final fusion time series shows the same changing during this period.

In section 3.1.2, line 179 page 7, 'Waveform classification is an effective method for identifying the noise observations' → please reformulate, as 'noise observations' is not clear. I guess you meant 'waveforms that are too noisy and difficult to process', do not you?

***Reply:***

  Thank you for your suggestion, the sentence has been revised into "Waveform classification is an effective method for identifying highly noisy waveforms that are challenging to process accurately."

Line 189 page 7, 'tracks with fewer than five valid observations were excluded from further analysis' → provide few words to justify why below 5 valid observations the track is discarded. Besides, the term 'valid' might not be the most appropriate. You are mentioning waveforms that does not fulfill the methodology shown in Figure 2 flow chart. It does not mean that kept waveforms will provide actual lake water level (see the case of the Mattenalpsee and Oberaarsee lakes with different value of the OLTC that you provided in your answer to one of my previous review comment).

***Reply:***

  Thank you for your suggestion, we revised the sentence into "After removing noisy observations, tracks containing fewer than five quality-controlled observations (post filtering via DEM-based elevation thresholds and waveform classification criteria) were excluded to ensure statistical reliability."

  Here is the quality-controlled by H_DEM and waveform classy, we would like to keep more points to make the two-step retracker more stable, so decide when smaller than 5 observations, the tracks would be excluded.

  OLTC shifts may introduce systematic elevation biases when two proximate lakes exhibit significant elevation differences ($\Delta H > 50$ m). In such cases, the tracking window could erroneously

lock onto the adjacent lake's surface, leading to implausible water level estimates for one of the pairs. This is not been processed in here, but later when do the cross validation with other satellite.

Figure 7, Change 'RMSE' with 'RMSE (in m)'

***Reply:***

Thank you for your suggestion, we update the figures in revised version.

In section 4.3 concerning 'Potential source of error', you should also discuss the impact of ice and snow on lake WSE estimate, which impacts a lot the radar altimeter signal. See the important literature on this issue.

***Reply:***

Thank you for your comment, the impact of ice and snow is big, we add several sentence to describe it. "Additionally, ice and snow cover introduce significant uncertainties in radar altimeter measurements. During frozen periods, radar pulses may penetrate snow/ice layers, measuring subsurface features rather than the true water surface (Guerreiro et al., 2017). Ice formation alters surface reflectivity, causing peak retracking misidentification (e.g., false peaks from ice-water interfaces; Chen et al., 2021, Song et al., 2020)."

In section 5, you should remove the lake without statistically significant trends in figures 8 to 10 and the numbers provided in section 5. You should write in the text that you computed the p-value and explain which threshold you set on p-value to assess if the trend is statistically significant or not. In appendix B, there are multiple lakes with p-value>0.01 and even some with p-value>0.05. You should also write in the text the number lakes without a statistically significant trend.

***Reply:***

Thank you for your suggestion, we have removed the lake without statistically significant trends through setting at the level 0f 0.05 (p<0.05) to assess if the trend is statistically significant or not, and also wrote in the text the number lakes with a statistically significant or not. Please see the revised manuscript.